

# The impact of ship emissions on air quality and human health in the Gothenburg area - Part I: 2012 emissions

Lin Tang[1,2], Martin O.P. Ramacher[3], Jana Moldanová[1], Volker Matthias[3], Matthias Karl[3], Lasse Johansson[4], Jukka-Pekka Jalkanen[4], Katarina Yaramenka[1], Armin Aulinger[3], Malin Gustafsson[1]

[1]IVL, Swedish Environmental Research Institute, P.O. Box 530 21, 40014 Gothenburg, Sweden
[2]WSP Environment Sweden, Box 13033, 402 51 Gothenburg, Sweden
[3]Chemistry Transport Modelling, Helmholtz-Zentrum Geesthacht, 21502, Geesthacht, Germany
[4]Finnish Meteorological Institute, P.O. Box 503, 00101 Helsinki, Finland

*Correspondence to*: Jana Moldanová (jana.modanova@ivl.se) and Lin Tang (lin.tang@wsp.com)





**Abstract.** Ship emissions in and around ports are of interest for urban air quality management in many harbour cities. We

investigated the impact of regional and local ship emissions on urban air quality for 2012-year conditions in the city of

Gothenburg, Sweden, the largest cargo port in Scandinavia. In order to assess the effects of ship emissions, a coupled regional

and local-scale model system has been set up, using ship emissions in the Baltic Sea and the North Sea, as well as in and

around the port of Gothenburg. Ship emissions are calculated with the Ship Traffic Emission Assessment Model (STEAM)

model taking into account individual vessel characteristics and vessel activity data. The calculated contributions from local

and regional shipping to local air pollution in Gothenburg were found substantial, especially in areas around the city ports. The

local shipping contribution of $NO_2$ to annual mean concentrations was up to 3.3 ppb, together with contribution from regional

shipping at the North Sea and the Baltic Sea, the contribution was up to 4.3 ppb. In an area close to the city terminals, the

contribution of $NO_2$ from local shipping was higher than that of the road traffic, which indicates importance of controlling the

local shipping emissions. The local shipping emissions of $NO_x$ decreased the summer mean $O_3$ levels in the city by 0.5 ppb on

annual mean. The regional shipping lead to a slight increase in the $O_3$ concentrations, however, the overall effect of the regional

and the local shipping together was a small decrease of the summer mean $O_3$ concentrations in the city. For $PM_{2.5}$, the local

ship emissions contributed with 0.1 µg m$^{-3}$ to the annual mean concentrations on the city-domain average, regional shipping

was under 2012 conditions a larger contributor to the local $PM_{2.5}$ than the local shipping, with an annual mean contribution of

0.5 µg m$^{-3}$ on the city-domain average.


Based on the modelled local and regional shipping contributions, the health effects of $PM_{2.5}$, $NO_2$ and ozone were assessed

using the ALPHA-RiskPoll (ARP) model. An effect of the shipping-associated $PM_{2.5}$ exposure in the modelled area was a

mean loss of the life expectancy by 0.015 years per person. The relative contribution of the local shipping to the impact of

total $PM_{2.5}$ was 2.2 % which can be compared to 5.3 % contribution from the local road traffic. The relative contribution of

the regional shipping was 10.3 %. The mortalities due to the exposure to $NO_2$ associated to shipping were calculated to be 2.6

premature deaths/year. The relative contribution of the local and the regional shipping to the total exposure to $NO_2$ in the

reference simulation was 14 % and 21 %, respectively. The shipping related ozone exposures were due to the NO titration

effect, leading to negative number of premature deaths. Our study show that overall health impacts of regional shipping can

be more important than those of local shipping, emphasising that abatement policy options on city-scale air pollution require

close cooperation across governance levels. Our findings indicate that the strengthened Sulphur Emission Control Areas

(SECA) fuel sulphur limit from 1 % to 0.1 % in 2015, leading to strong decrease in formation of secondary particulate matter

on regional scale, has been an important step in improving of the air quality in the city.




**Copyright**

The works published in this journal are distributed under the Creative Commons Attribution 4.0 License. This licence does
not affect the Crown copyright work, which is re-usable under the Open Government Licence (OGL). The Creative
Commons Attribution 4.0 License and the OGL are interoperable and do not conflict with, reduce or limit each other.
© Crown copyright 2020









# 1 Introduction

Shipping is an important source of air pollutants, both on the global and European level (Corbett et al., 1999; Eyring et al., 2005; Cofala et al., 2007). The most important species emitted are sulphur oxides ($SO_x$), nitrogen oxides ($NO_x$), particulate matter (PM) and to some extent carbon monoxide (CO) and volatile organic compounds (VOC). Since nearly 70 % of ship emissions occur within 400 km of coastlines (Corbett et al., 1999), the largest contributions of shipping to air pollution are concentrated to coastal regions with intensive ship traffic and to harbours, where emissions from harbour operations add further to the air pollution generated by ships. The primary air pollutants from shipping contribute to the formation of secondary air pollutants, mainly ozone and secondary particulate matter. On average shipping emissions contributed with 9.4 % to concentrations of primary $PM_{2.5}$ (particulate matter with a median aerodynamic diameter less than or equal to 2.5 µm) and with 12.3 % to concentration of secondary inorganic particulate matter over the Europe during 1997–2003 (Andersson et al., 2009).

Emissions from the international shipping are controlled through International Maritime Organisation (IMO) and regulations included in the International Convention on the Prevention of Pollution from Ships (MARPOL 73/78) and its annexes. The MARPOL Annex VI- "Regulations for the Prevention of Air Pollution from Ships" sets limits for emissions of $SO_x$ and $NO_x$. Sulphur is regulated through maximum allowed sulphur content in the fuel used, while $NO_x$ is regulated through Tier limits for maximum specific emissions of $NO_x$ from each engine on board. The limits depend on the nominal rotation speed of an engine and different Tiers apply for ships built or substantially re-built in different time periods (2000–2011 Tier 1, after 2011 Tier 2). For fuel sulphur content (FSC) a global limit of 0.5 % applies since 1st January 2020, before it was 3.5 %. However, the Baltic Sea, the North Sea and the English Channel are so called Sulphur Emission Control Areas (SECA) where stringer limits apply: in July 2010 it was decreased from 1.5 % to 1.0 %, which is also the limit that applies in this study. In 2015 the fuel sulphur limit was decreased further to 0.10 %. In addition, since 2010 a sulphur content limit of 0.10 % for fuels used by ships at berth for a period longer than 2 hours applied for all EU ports. Sweden has also introduced economic incentives for reduction of the shipping emissions in form of differentiated fairway and port fees with a discount for ships using emission control technologies, contributing to a relatively large share of ships with $NO_X$ abatement technology in the region. In 2020 the global cap for the FSC will be decreased to 0.50 %. In 2021 a Nitrogen Emission Control Area (NECA) will enter in force in this area with mandatory Tier 3 standard (80 % reduction comparing to Tier 1) for ships built in 2021 and later operating in the region.

In the Baltic Sea and the North Sea, an intensive ship traffic results in high emissions of air pollutants, and contributes to high atmospheric concentrations of particularly of $NO_x$ in and around several major ports (Jonson et al., 2015). The relative contribution of shipping in the North Sea and Baltic Sea to coastal $NO_2$ concentrations are highest along the coasts of southern Sweden, the south-western coast of Finland, and the coast of Estonia, 25–40 % on annual average (Jonson et al, 2019). Jonson et al. (2019) found that the Baltic Sea and North Sea shipping contributed significantly also to concentrations of particulate





matter (highest contributions 6–12 %, allocated to similar areas as $NO_x$) and to the deposition of sulphur (highest contributions

10–20 %), before the strengthening of SECA fuel sulphur limit. They have also shown that the strengthened limit on the fuel

S content in 2015 from 1.0 % to 0.10 % brought a significant decrease in emissions as well as contributions of shipping to air

pollution by $SO_2$ and to S deposition (maximum contribution about 2 %) and to a reduction of shipping contribution to the

concentrations of PM. Aulinger et al. (2016) and Matthias et al. (2016) studied impacts of the current and future (2030) North

Sea shipping on air pollution and found contributions consistent with Jonson et al. (2019) (highest $NO_2$ contributions 25 %

and 15 % in summer and winter, respectively, ozone increased by 10% along Scandinavian coast). By 2030, the contribution

of shipping to the $NO_2$ and $O_3$ concentrations was estimated to increase by more than 20 % and 5 %, respectively, due to the

expected enhanced traffic, if no regulation for further emission reductions is implemented in the North Sea area (Matthias et

al., 2016).


Several studies have assessed impacts of shipping on human exposure to air pollutants and associated to health impacts.

Andersson et al. (2009) assessed impacts of different source regions and also of emissions from international shipping on

personal exposure and to the relative increase of death rates from exposure to particulate matter across Europe with help of the

atmospheric chemistry-transport model MATCH. They found that shipping, before the introduction of a SECA in the region,

contributed with 5 % to population-weighted average concentration (PWC) of primary $PM_{2.5}$ and with 9 % to PWC of

secondary inorganic particles. For individual countries in Northern Europe the contribution to PM exposure varied between 3

% and 19 %. Jonson et al. (2015) assessed health impacts of $PM_{2.5}$ associated to emissions from ships in the Baltic Sea and the

North Sea in years 2009 and 2011, i.e. before and after the SECA FSC limit was strengthened from 1.5 % to 1.0 %, with help

of EMEP chemistry-transport model. The relative contributions of shipping to population exposure to $PM_{2.5}$ were found

between 1.6 % and 12 % for 2009, and 1.4 % and 10 % for 2011 for the riparian countries, decreasing by 0–40 % between

these years in the different countries. Contributions from shipping to the total exposure to particles in these countries found by

Jonson et al. (2015) for year 2009 were by 14–64 % lower than those found in Andersson et al. (2015), accounting that, apart

from differences in models and meteorological years used in the 2 studies, Andersson et al. assessed impact of all European

shipping prior to SECA regulation entered into force while Jonson et al. assessed impact of the Baltic Sea and the North Sea

shipping after the introduction of the 1.5 % SECA fuel sulphur content limit. Barregård et al. (2019) assessed impact of

shipping in the Baltic Sea for emission years 2014 and 2016, i.e. before and after strengthening of SECA FSC limit from 1.0

% to 0.1 % using the EMEP model and showed that exposure to $PM_{2.5}$ associated to the Baltic Sea shipping decreased by 34

% in the region due to this abatement measure, using emissions representative for year 2016, shipping contributed with 10 %

to the population exposure of $PM_{2.5}$ in the coastal regions but only less than 1 % in more remote inland areas.


The methodologies for calculation of the health impacts of $PM_{2.5}$ in the above discussed studies vary both in the

exposureresponse functions (ERF) used and how the years of life lost are calculated from statistics of mortalities and life-





tables. The most common ERF used is the one recommended by the HARPIE study (WHO, 2013a), increased risk of all-cause mortality 1.0062 (95 % CI 1.004–1.008) per µg/m$^3$, which is almost the same as ERF from Poppe et al. (2002). Several studies

use a higher ERF presented in Jerret et al. (2005) and in the ESCAPE study (Beelen et al., 2014), both of very similar value, the latter being 1.014 (95 % CI 1.004–1.026) per µg/m$^3$. Andersson et al. (2009) calculated increase of death rates from exposure to particulate matter in Europe using ERF from Poppe et al. (2002) for the secondary inorganic aerosol and ERF from Jerret et al. (2005) for the primary $PM_{2.5}$, reasoning that the ERF of Jerret et al., based on intra-city gradients is better representing the impact of primary $PM_{2.5}$, while Poppe et al. uses the inter-city differences reflecting more impact of secondary

PM. Combining the increase of mortality from particulate matter in EU27 and the relative contribution of shipping to the exposure to primary and secondary inorganic PM, Andersson et al. (2015) found the resulting impact of shipping on mortality 22 000 premature deaths per year. Jonsson et al. (2015) used the RAINS methodology which calculates years of life lost (YOLLs) over the expected lifetime of population in risk, in this case population above 30 years, accumulating YOLLs between the ages of 30 and c.a. 80 years (Amann et al., 2004). The RAINS methodology uses ERF recommended by the HARPIE

project. As a result, Jonson et al. (2015) estimated 0.1–0.2 YOLLs per person in areas close to the major ship tracks resulting from the ship emissions in the Baltic Sea and the North Sea for year 2010. Barregård et al. (2019) estimated that 187-421 premature deaths per year, corresponding to 0.01–0.02 YOLLs per person, could be associated with contributions of the Baltic Sea shipping emissions to concentrations of $PM_{2.5}$ in year 2014. The lower and higher estimates used ERF from WHO (2013) and Beelen et al. (2014), respectively. In our study the impacts of exposure to shipping-related air pollutants on health of

people living in the Gothenburg region have been assessed using the ALPHA-RiskPoll methodology (ARP, Holland et al. 2013, Åström et al., 2018) which uses the ERFs from the HARPIE project (WHO, 2013a).

The city of Gothenburg is located on the western coast of Sweden, with about 0.57 million inhabitants and an area of 450 km$^2$. The dominant wind direction in Gothenburg is south-west with average wind speed of 3.5 m s$^{-1}$, indicating the major transport

path from sea to the land, especially in summer. The geomorphology of the Gothenburg area is described as a fissure valley landscape dominated by a few large valleys in north-south and east-west directions. The major air pollution sources in Gothenburg are above all road traffic and industry, wood burning, shipping, agriculture, working machines and long-range transport (LRT) from the European continent and other parts of Sweden. The harbour and shipping activities are important emission sources and directly influences the urban air quality. The centre of the city is situated on the southern shore of the

river Göta älv. The Port of Gothenburg receives between 6,000 and 6,500 calls per year and additional 600–700 ships pass to and from ports upstream and on the Göta älv. The port annually handles approximately 900,000 containers, 20 million tonnes of petroleum, and half a million Roll-on/roll-off (RoRo) units (Winnes et al., 2015). Passenger traffic in Gothenburg is also very busy with 1.5 million passengers who ferry to and from Gothenburg to Denmark, Germany *etc.* on Stena Line ferries each year. This makes the port the largest cargo port in Scandinavia.




Comparing with other European cities, the air pollution levels in Gothenburg are low and the air quality has become better and better since the 70s because of the effective emission control addressing industry and road traffic. The trends of $SO_2$, $NO_x$ and $NO_2$ have been continuously decreasing from 1990 to 2015, except for the areas close to major roads (Miljöförvaltningen, 2017). $O_3$ exhibits an increasing trend and there is also a slowly increasing trend for $PM_{10}$ in Gothenburg (Olstrup et al., 2018).

The annual means for $NO_2$, $PM_{10}$ and $PM_{2.5}$ (particulate matter with aerodynamic diameter less than or equal to 10 µm and 2.5 µm, respectively) during the period 2000–2017 are 12.5 ppb, 16.3 µg/m$^3$ and 7.9 µg/m$^3$, respectively, at an urban background site in Gothenburg. The decreased levels of $NO_x$ and $NO_2$ during the period 1990–2015 in Gothenburg were estimated to increase the life expectancy by up to 12 months and 6 months respectively, and the slight increased trend of $O_3$ and $PM_{10}$ have relatively little impact on life expectancy (-2 month and -1 month) (Olstrup et al., 2018). In terms of exposure to $PM_{10}$ and

$PM_{2.5}$ from different source categories in Gothenburg, Segersson et al. (2017) calculated that the largest part was due to the long-range transport and the dominating local sources were road traffic and residential wood combustion, while the contribution from local shipping was small, 0.04 µg/m$^3$ population weighted annual mean $PM_{2.5}$. The exposure of $PM_{2.5}$ from shipping in other harbour cities in Sweden are lower than in Gothenburg, with 0.02 µg/m$^3$ in Stockholm and 0.01 µg/ m$^3$ in Umea (Segersson et al., 2017).


This study has been conducted within the BONUS SHEBA project (Shipping and Environment of the Baltic Sea Region) where the impact of current and scenario emissions from ships on air quality have been investigated as a part of a holistic assessment framework for impacts of shipping on marine and coastal environment. The shipping-related air pollution has been investigated on a range of spatial scales with several chemistry-transport models: coarse spatial scale resolution was used for

simulations in the European domain, finer resolution was used for the Baltic Sea (Karl et al., 2019a, c), and city-scale simulations using high spatial resolution were used for several harbour cities (Ramacher et al., 2019a). The present study (Part I) evaluates the contributions of regional and local shipping to the concentrations of $SO_2$, $NO_2$, $PM_{2.5}$, $O_3$ and secondary PM, as well as the human exposure and the associated health impacts in Gothenburg for year 2012. Health impact studies for the shipping emissions in cities are rare, mainly because the spatial resolution of the regional CTM (Chemical Transport Model)

does not allow for the city-scale resolution. This study provides the city-scale Health Impact Assessment (HIA) and identifies and addresses potential health impacts associated to local and regional shipping. The studied year (2012) has been considered as a present-day "normal year" for Baltic Sea Region in terms of meteorological conditions in BONUS-SHEBA. In terms of ship emission regulations, the study presents a situation with 3.5 % FSC global limit, 1.0 % FSC limit in the SECA area whereas 0.1 % FSC limit applies for ships berthing in the port of Gothenburg or operating within the Göta älv estuary. Several

alternative shipping scenarios in year 2040 are discussed further in Ramacher et al., 2019b (Part II).



## 2 Methodology

### 2.1 Model set-up

For the city-scale chemistry transport model (CTM), the prognostic meteorology-dispersion model TAPM (The Air Pollution Model) (Hurley et al., 2005; Hurley, 2008) was used. TAPM consists of a meteorological component and an air quality component. The meteorological component of TAPM is an incompressible, non-hydrostatic, primitive equation model with a terrain-following vertical sigma coordinate for 3-D simulations. Using predicted meteorology and turbulence from the meteorological component, the air pollution component solves prognostic equations for concentrations and cross-correlation of concentrations by the Eulerian grid module. It includes gas-phase photochemistry based on the Generic Reaction Set (Azzi et al., 1992), gas- and aqueous-phase chemical reactions for $SO_2$, formation of ozone from $NO_X$ and NMVOC (treated as VOC reactivity) and simple formation of secondary inorganic and organic aerosol. The model treats also dry and wet deposition processes of gases and PM.

In this study, the meteorological component of TAPM was driven by the recently published ECMWF ERA5 synoptic meteorological reanalysis ensemble means with 30 vertical layers, $0.3° \times 0.3°$ horizontal and three-hour temporal resolution. For a time period of 2012, five nested domains have been simulated with the synoptic meteorological component with the inner-most meteorological fields of a 30 km × 30 km domain with 500 m resolution. In addition, the observed wind fields at four meteorological sites were assimilated to nudge wind speed and wind direction calculations in the inner-most domain.

The spatial resolution of the city-scale CTM was 250 m × 250 m with local coordinate system SWEREF 99 1200 and the size of the CTM domain was about 25 km × 25 km, covering the city of Gothenburg and the harbour area along shores of the Göta älv running through the city (Fig. 3).

The chemical boundary conditions were taken from the Community Multi-scale Air Quality Modelling System (CMAQ) (Byun and Ching, 1999; Byun and Schere, 2006). CMAQ model simulations on a 4 km × 4 km grid (Karl et al., 2019c), which were used for the chemical boundary conditions, were driven by the high-resolution meteorology meteorological fields of the COSMO-CLM (Rockel et al., 2008) version 5.0 using the ERA-Interim re-analysis as forcing data. Chemical boundary conditions for the CMAQ model simulations were provided through hemispheric CTM simulations, from a SILAM model (Sofiev et al., 2006) run on a $0.5° \times 0.5°$ grid resolution, which was provided by Finnish Meteorological Institute (FMI). Land based emissions for the regional-scale simulations were represented by hourly gridded emissions calculated with SMOKE-EU emission model (Bieser et al., 2011). The SMOKE-EU emission data is based on reported annual total emissions from the European point source emission register (EPER), the official EMEP ([www.ceip.at](www.ceip.at)) emission inventory and the EDGAR HTAP v2 database (EPER, 2018; CEIP, 2018; Olivier et al., 1999). For shipping emissions, the model used an inventory calculated with the STEAM model consistent with the inventory used by TAPM for the city-scale, calculated with 2 km × 2 km grid





resolution (STEAM3, Johansson et al., 2017), more details are given in the next section. The STEAM model version used for

the CMAQ simulations was, however, not including VOC emissions. As chemical boundary conditions, vertical model layer

seven with a mid-layer height of approximately 385 m above ground was selected. CMAQ simulations with and without ship

emissions in the Baltic Sea and the North Sea included were used in TAPM simulation runs. Since the TAPM allows just 1-d

boundary concentration fields with hourly time resolution, the TAPM boundary concentrations were calculated using the

horizontal wind components on each of the four lateral boundaries for weighting the upwind boundary concentrations around

the TAPM model domain (Fridell et al., 2014). The city-scale model set-up are summarised in Table 1.

### 2.2 Emission inventory

### 2.1.1 Regional and local shipping emissions

Shipping emissions were calculated with the Ship Traffic Emission Assessment Model taking into account individual vessel

characteristics and vessel activity data (STEAM; Jalkanen et al., 2012; Johansson et al., 2017) based on detailed information

of technical parameters of individual vessels and position data of individual ships taken from reports from the Automatic

Identification System (AIS) of the Helsinki Convention (HELCOM) member states. The STEAM model calculated fuel

consumption and emissions as a function of vessel activity, the STEAM3 version of the model has been used (Johansson et

al., 2017), with an additional module for calculation of VOC emissions. The emission inventory includes combustion emissions

from all engines and appliances on ships (boilers, auxiliary and main engines). The emission inventory for the local shipping

around Gothenburg consists of hourly emissions from ships on 250 m × 250 m grid resolution. The STEAM model provided

shipping emissions for year 2012 for the compounds $NO_x$, $SO_x$, CO, $CO_2$, VOC and $PM_{2.5}$. $PM_{2.5}$ is further divided into

Elemental Carbon (EC), Organic Carbon (OC), $SO_4^{2-}$ and mineral ash. Ship emissions were provided in two vertical layers

with emissions below and above 36 m height in order to differentiate between emissions from large ships with high stacks and

the smaller vessels with lower stack heights (Fig. 1). The stack release heights were attributed to the corresponding mid-points

of model layers in TAPM, 15 m for the emissions below 36 m height and layer 36 m for emission above 36 m.

### 2.2.2 Road traffic emissions

The road traffic emissions were calculated from traffic activity data and emission factors. The basic set of emission factors

from road vehicles was extracted from HBEFA v. 3.2 (HandBook Emission FActors for Road Transport, Rexeis et al., 2013).

HBEFA comprehends emission factors for different classes of road vehicles based on type of vehicle (e.g. motorcycles, light-

duty, heavy-duty vehicles), technology or fuel (e.g. petrol, diesel, hybrid) and emission standard (pre-Euro–Euro 6). For each

of those also a number of road categories and driving patterns that affect emissions are specified within the vehicle sub-

segments. The emission factors for light-duty and heavy-duty vehicles and busses in Gothenburg were calculated using the

Swedish national database on car fleet composition and national, vehicle-type specific, activity data in 2012. Road traffic

emissions were finally calculated using traffic activity data for Gothenburg (vehicle kilometres for light duty vehicles plus

motorcycles, heavy duty vehicles and busses on road links with specified type, speed and congestion hours) from the database





of the Environmental Administration, City of Gothenburg (Miljöförvatningen), and corresponding emission factors calculated in the HBEFA database. These data were applied as line emission sources in the model.

### 2.2.3 Other emissions

Ten large point sources from industrial processes are present in the city-scale model domain, these are all fugitive emissions from fuel handling and refineries. For technical reasons these were considered as area sources in the model with release heights corresponding to the stack heights allocated to these sources. The emission factors from these industrial sources were obtained from Swedish Environmental Emission Data (SMED 2015) for 2012.

Emissions from the following sectors were geographically distributed on 1 km × 1 km grid and assigned with coordinates and emission height: 'Manufacture of solid fuels and other energy industries', 'Combustion in industry for energy purposes', 'Stationary combustion in agriculture/forestry/fisheries', 'Energy and heat production (commercial/institutional)', 'Residential plants (boilers), domestic heating, working and off-road machinery', 'Use of paints and chemical products in households and enterprises', 'Agriculture, waste and sewage', as well as 'Other transports' (the landing and take-off emissions from aviation, trains and military). They stem also from the SMED database and were obtained from SMHI (Swedish Meteorological and Hydrological Institute). These emissions were applied as gridded sources in the model.

The local emissions of $NO_x$, $SO_2$, $PM_{10}$ and VOC from the above-mentioned sectors in the model domain are shown in Fig. 2. The local shipping was the dominant emission source of $SO_2$ contributing with 61 % to the total $SO_2$ emissions (502 ton/year) in the model domain. Further, local shipping contributed with 41 %, of the $NO_x$ emissions in the model domain, which was comparable to contribution from the road traffic (47 %), to total NOX emissions of 5072 ton/year. However, the road traffic was the major contributor for $PM_{10}$ in the model domain (45 % of 357 ton/year), while the local shipping and industry contributed with approximately 25% each. For VOC (7457 ton/year), about 92 % of emissions were released from the industrial sector and only 0.3 % from the local shipping.

### 2.3 Design of model simulations

Several model simulations were performed to investigate the influence of shipping within the city domain and influence of the regional shipping outside the city on air pollution in 2012:

(1) A simulation including complete emission inventory both in the city-scale simulation and in the CMAQ simulation supplying the chemical boundary conditions: "Base scenario";

(2) A simulation excluding the local shipping in the TAPM city domain but including regional shipping chemical boundary conditions in the CMAQ simulation: "No local shipping scenario" and;

(3) A simulation excluding both local shipping in the TAPM city domain and shipping in the chemical boundary conditions in the CMAQ simulation: "No local and regional shipping scenario".






In addition, three sensitivity studies were performed within this study:

(1). "No NMVOC from local shipping" had the same emission input as the "Base scenario", but without NMVOC from local shipping emissions. The difference between "Base" and "No NMVOC from local shipping" was used to investigate the impact of VOC emissions from local shipping, which is often neglected in the emission inventories due to its small

proportion and as these has only recently been included in the STEAM model;

(2). "Primary PM from local shipping" had the same emission input as the "Base scenario", but for the local shipping only the primary PM emissions as calculated by the STEAM model were included, all emissions of the gaseous species were excluded preventing formation of the secondary PM from local shipping. The difference between "Base scenario" and "Primary PM from local shipping" reflects the formation of secondary PM from $SO_2$, $NO_x$ and VOC emitted by the local

shipping;

(3). "No road traffic" had the same emissions as the "Base scenario", but without road traffic emissions. It was used to compare the contributions of shipping emissions as well as the health impact of shipping with emissions from the city traffic.

## 2.4 Model evaluation

Model evaluations were carried out for both meteorological and air pollution parameters. The simulated meteorological parameters (temperature, relative humidity, wind fields and precipitation) were evaluated with measurements at four stations: Femman (57.70° N, 11.97° E, 30 m a.s.l.), Göteborg A (57.72° N, 11.99° E, 3 m a.s.l.), Vinga A (57.63° N, 11.60° E, 18 m a.s.l.) and Landvetter (57.68° N, 12.29° E, 154 m a.s.l.). The urban background site Femman is located on a rooftop in the city center and the local meteorological variables as well as the air quality data are continuously measured by the Environmental

Administration in Gothenburg (Miljöförvaltningen), while the other three meteorological stations are driven by SMHI. For the air pollution evaluation, Femman, Haga and Mölndal were included. Haga is located in a one-sided street canyon in central Gothenburg with a park to the east of the station. Mölndal is located in southern part of Gothenburg on a rooftop about 30 m above ground and corresponds to a traffic station. $NO_2$ is measured by a reference chemiluminescent method at Femman and Haga and by Differential Optical Absorption Spectrometry (DOAS) method at Mölndal. $PM_{10}$ mass concentrations are

measured by TEOM (Tapered Element Oscillating Microbalance, model 1400ab) instrument at all three stations. Ozone instrument at Femman was Teledyne (model T400) and at Mölndal a DOAS from OPSIS. Hourly averaged air quality data for $NO_2$, $PM_{10}$ and $O_3$ at the three air quality stations were used to evaluate the model performance.

The FAIRMODE DELTA Tool version 5.4 was used for the evaluation of the model results for the city of Gothenburg. DELTA

Tool is an IDL (Interface Definition Language)-based statistical evaluation software which allows to perform diagnostics of air quality and meteorological model performance (Thunis et al., 2012, Pernigotti et al., 2014). The tool focuses on the air pollutants regulated in the Air Quality Directive 2008 (AQD 2008) and calculates statistical performance indicators such as





Mean, Exceed, Normalized Mean Bias (NMB), Normalized Mean Standard Deviation (NMSD) and High Percentile ($H_{perc}$).
Moreover, a performance criteria can be calculated, which is combining the statistical performance indicators with fixed
parameters to evaluate whether the model results have reached a sufficient level of quality for a given policy support application
(Pernigotti et al. 2014). According to the DELTA tool, the capability of a model to reproduce measured concentrations is good
when more than 90 % of the stations fulfil the performance criteria. We applied the Delta Tool to concentrations of $NO_2$, $PM_{10}$
and $O_3$ measured at the available urban background sites and road traffic sites, compared them with concentrations calculated
by our model system and calculated both, statistical performance indicators and the model performance criteria.

## 2.5 Health impact assessment

The health impacts of exposure of population in Gothenburg to shipping-related air pollutants were assessed with the ALPHA-
RiskPoll (ARP) methodology (Holland et al., 2013) providing for calculation of a wide range of air-pollutant specific health
effects based on the population weighted concentrations, national population statistics on age distribution of population,
mortality and morbidity data and effect-specific exposure-response relationships. The methodology has been developed and
used for quantification and assessments of the benefits of air pollution controls in Europe for the UN/ECE Convention on Long
Range Transport of Air Pollution and is based on work for the Clean Air For Europe (CAFÉ) Program and on EU project
Modelling of Air Pollution and Climate Strategies (EC4MACS). Following the WHO recommendations (WHO 2013a) and
the Clean Air for Europe (CAFÉ) cost-benefit analysis methodology for assessment of health impacts of air pollutants, impacts
of exposure to $PM_{2.5}$, ozone and $NO_2$ have been considered in the analysis. The exposure to the three pollutants are considered
most harmful by the World Health Organization (WHO, 2013b). In this study only the most serious impacts, i.e. losses of lives,
are presented, taking into account impacts of long-term exposure to $PM_{2.5}$ and short-term exposures to ozone and $NO_2$, i.e. the
impacts marked A* in the HARPIE study (WHO 2013a). For ozone the indicator SOMO35 is used, representing the 8-hourly
mean ozone concentrations accumulated dose above a threshold of 35 ppb during a year. The health impacts of some pollutants
are correlated and that is why the premature deaths attributed to each pollutant cannot simply be added up. In particular, it has
been estimated that adding premature deaths attributed to $PM_{2.5}$ to those attributed to $NO_2$ could result in double counting of
around 30 % (WHO 2013a). The health impacts calculated with the ARP model are presented as premature deaths and YOLLs
per year, using the ER function of the model, i.e. 1.0062 (95 % CI 1.004–1.008) per $\mu g/m^3$ (WHO, 2013a).

The concentrations fields of $PM_{2.5}$, $O_3$ and $NO_2$ were calculated by the coupled high resolution (250 m × 250 m) modeling
system, as described above. Annual means and SOMO35 were calculated from hourly concentrations for each grid. Population
data on 1 km × 1 km resolution were obtained from Statistics Sweden (SCB) for 2015 with a population of 572 779 in the city
of Gothenburg. As there are no significant changes in population density between 2012 and 2015, the population data for 2015
were used. Population-weighted average concentrations (PWC) for the model domain were calculated multiplying the
modelled annual mean concentration of the pollutant on each grid-cell by the population in the same grid-cell as weight for
the modelled concentration.



To calculate the health risks, further information needed is the ERF and the baseline health statistics including the life expectancies, the death rates and morbidity data for estimating the impacts on mortality and morbidity. To estimate YOLLs, the age at which the premature deaths occurred should also be considered. In the ARP model, the ERFs used are those from
WHO (2013a): 6.2% (95% confidence interval 4.0–8.3%) relative risk increase per 10 µg/m$^3$ increased exposure for the PM$_{2.5}$ exposure, 0.29% (95% confidence interval 0.14–0.43%) relative risk increase per 10 µg/m$^3$ increased exposure for the ozone exposure and 0.27% (95% confidence interval 0.16–0.38%) relative risk increase per 10 µg/m$^3$ increased exposure for the NO$_2$ exposure. The YOLLs are calculated per year. The analysis was made separately for the population exposure related to the different pollutants from local and regional shipping.

## 3 Results and discussion

### 3.1 Model evaluation

The model evaluation was conducted for both meteorology and air pollution in the inner-most model domain. The comparison between measured and modelled local meteorological parameters (temperature, relative humidity, total solar radiation, wind speed, wind direction and precipitation) shows high correlation and low bias. The application of ERA5 datasets in the model
shows significant improvements from the default reanalysis datasets. Nevertheless, the predictions of the meteorological parameters such as wind fields flow get better with wind field assimilation, for more detail see Ramacher (2018). For example, the differences between observed and simulated wind rose at Femman in January and July indicate a good model capability to reproduce local wind field except for missing about 30% of low wind speeds (0–2.5 m s$^{-1}$) from the north (Fig. 4), which may introduce some underestimations in high pollutant concentrations at ground due to accumulation in the boundary layer.
Nevertheless, the total frequency of northerly winds at Femman station is low in January (8–17%) and very low in July (1–8%).

The evaluation of ambient pollutants was conducted through the major statistical parameters. At urban background site, the estimation of NO$_2$ and PM$_{10}$ concentrations were satisfactory in summer with lower bias, however the model tended to
underestimate NO$_2$ and PM$_{10}$ concentrations in winter. O$_3$ evaluation was carried out at station Femman and Möldal, and underestimation of daily maximum of the 8-hour means was also detected, which could be caused by low resolution of local NO sources and hence more smoothed titration of ozone. The summary statistics according to the FAIRMODE model evaluation tool shows that less than 90% of daily PM$_{10}$ concentrations at road site Haga fulfill the performance criteria for the statistic indicator H$_{perc}$ (Fig. 5). The indicator H$_{perc}$ indicates the model capability to reproduce extreme events, represented by
selected high percentile for modelled and observed values.



The underestimation of $NO_2$ and $PM_{10}$ especially at road sites demonstrate impact of too coarse spatial resolution (250 m ×
250 m) not capturing high concentrations at street level, possible missing or insufficient cover of local emissions like
resuspension particulate matters from traffic sources, or incomplete chemical reactions in the model etc. As pointed out by

Karl (2019b), recent nested model approaches have not resolved the details in the emission processing and near-field dispersion
at the street and neighborhood level. However, shipping emissions are, when reaching the exposed population, more dispersed
and the 250 × 250 $m^2$ grid resolution should be sufficient to assess their impact. Nevertheless, the other statistic indicators
(Mean, exceedances, Normalized Mean Bias, Normalized Mean Standard Deviation, Correlation coefficient, etc.) of model
performance in Fig. 5 show a satisfactory performance of the used city-scale model for Gothenburg.

**3.2 Impact of ship emissions on local air quality**

**3.2.1 $SO_2$**

The modelled annual average $SO_2$ concentrations from all sources, including local and regional shipping are shown in Fig. 6.
The calculated annual mean concentration of $SO_2$ from all sources in the model domain is 0.4 ppb and local shipping
contributes 0.05 ppb on model-domain average and up to 0.6 ppb in a wide area around the main shipping routes and ports.

An additional increase of 0.1 ppb $SO_2$ is detected when considering the regional ship emissions. In summer months (JJA), local
and regional shipping contribute on model-domain average with 0.1 ppb to $SO_2$ concentrations and with 0.8 ppb in maximum
(Fig. S1 in the Supplement). The highest $SO_2$ contributions (maximum 0.6 ppb on annual mean) were found around the major
ports: Älvsborgshamnen, Skandiahamnen, Skarvikshamnen, Ryahamnen, Lindholmshamnen, and Frihamnen in the northern
bank of the Göta älv (Fig. 6d). In addition, two busy ferry terminals located on the southern bank of the Göta älv can contribute

to the high $SO_2$ concentrations on the opposite river side due to the dominant south-westerly winds.

**3.2.2 $NO_2$**

$NO_x$ is mainly emitted as nitrogen oxide (NO), in the STEAM model the $NO_2/NO_x$ ratio is 5%. In atmosphere NO is quickly
converted to $NO_2$ in reaction with ozone, so further from the source the atmospheric $NO_x$ is dominated by $NO_2$, approaching
a photo-stationary state driven by the $NO+O_3$ reaction and $NO_2$ photolysis. Maps of modelled annual mean atmospheric

concentrations of $NO_2$ over the Gothenburg area are shown in Fig. 7. The annual mean concentration of $NO_2$ in the Base
simulation is 3.7 ppb as the model-domain average (Fig. 7a), and the model-domain mean contribution from local shipping to
the annual mean concentrations is 0.5 ppb and up to 3.3 ppb in areas with high contribution (Fig. 7b). The calculated model-
domain mean contribution to $NO_2$ concentrations from local and regional shipping together is 1.5 ppb (41%) and up to 4.3 ppb
in most heavily impacted areas (Fig. 7c). The seasonal differences in $NO_2$ concentrations are driven by emissions, atmospheric

chemistry and atmospheric mixing. Maps of modelled air concentrations of $NO_2$ over the Gothenburg area in winter and
summer month are shown in Fig. S2 in the Supplement. The higher contribution of local shipping in summer and larger
influenced areas is due to 20% higher summer emissions comparing to winter, different photochemical state as well as different





local meteorological conditions. The dominated south-westerly winds in summer transport $NO_2$ from the shipping routes and port areas farther inlands. Again, the highest level of $NO_2$ is around the port Skandiahamnen.


Eriksberg is a modern residential and commercial center built in place of an old dockyard area. The daily influence from local and regional shipping is obvious and illustrated by the calculated daily mean $NO_2$ concentrations from the three model simulations "Base", "No local shipping" and "No local and regional shipping" (Fig. 8). The local shipping contributes with 2.5 ppb to daily mean concentrations of $NO_2$ on average, and up to 11.1 ppb in March. Meanwhile the regional shipping

contributes with 1.0 ppb to daily mean $NO_2$ concentrations on average and reaches 9.2 ppb at most. For comparison, the daily average contributions of $NO_2$ from road traffic are also presented in Fig. 8. In total, 219 days in 2012 show contributions to daily mean concentrations from local shipping higher than those from road traffic at Eriksberg. Even though road traffic is the major contributor to the $NO_2$ concentrations in urban environment, the local ship emissions should not be neglected, especially in areas close to the city ports.

**3.2.3 $O_3$**

$O_3$ is formed in photocatalytic cycles involving NOx, ozone and hydrocarbons, through the photolysis of $NO_2$ in sunlight. The same cycle involves also titration of ozone by the reaction with NO forming the $NO_2$. Maps of modelled atmospheric concentrations of ozone over the Gothenburg area in 2012 are shown in Fig. 9, with focus on summer months (JJA). The regional background concentration of ozone at a regional background stations close to Gothenburg area is 37 ppb in the summer

of 2012. Modelled summer ozone levels in the model domain are in the 15–30 ppb range (Fig. 9a). Since $NO_x$ is mainly emitted as NO, the emissions from local shipping cause local reduction of ozone concentrations due to the titration of $O_3$ by NO in the main shipping routes and port areas (Fig. 9b). The $O_3$ depletion pattern along the north of the Göta älv is 4 ppb in maximum due to the local shipping with both NOx and NMVOC emissions (Fig. 9b), while regional shipping emissions tend to increase the ozone concentrations by 1 ppb over the land compared to 4–6 ppb over the remote ocean due to large-scale ozone

production in summer (Huszar et al., 2010, Fig. 9c).

In the local STEAM inventory, the non-methane volatile organic compounds (NMVOC) from shipping are also available. These NMVOC serve as precursors of $O_3$ and enhance photochemical ozone production. TAPM model uses concept of VOC reactivity instead of individual NMVOCs, producing pool of peroxy radicals which take part in the ozone-production

photocatalytic cycle. A sensitivity run was performed to study the impact of NMVOC emissions from local ships on ozone concentrations in the city by excluding the local shipping NMVOC emissions from the simulation. Fig. 9d shows the impact of the NMVOCs emissions: the $O_3$ concentrations increased by up to 2 ppb along the main shipping routes and the port areas, which means the negative effects of $NO_x$ emissions from local shipping on the ozone concentrations was 6 ppb in maximum when NMHC emissions were excluded comparing to 4 ppb in the Base simulation.





### 3.2.4 Particulate matter

Particulate matter includes primary, directly emitted particles and secondary particulate matter formed upon further processing of emissions in the atmosphere. At the urban background site Femman, close to the city harbour, the measured annual mean $PM_{2.5}$ concentration was 7.9 µg m$^{-3}$ in 2012. The calculated annual mean $PM_{2.5}$ in the Base simulation was 4 µg m$^{-3}$ as the model domain average (Fig. 10a). The local ship emissions contributed with 0.1 µg m$^{-3}$ (3 %) to the annual mean as the model-domain average (Fig. 10b). Regional shipping was under 2012 conditions a larger contributor to the local $PM_{2.5}$ than the local shipping with annual mean average contribution of 0.4 µg m$^{-3}$ (11 %) (Fig. 10c). In summer, the area with major influence of shipping emissions extended to the north of the Göta älv with maximum contributions from local plus regional shipping of 1.4 µg m$^{-3}$ at Skandiahamnen (A3 in Appendix). At the near-harbour residential area Eriksberg, the contribution from the local and regional shipping was in range 0.2–1.1 µg m$^{-3}$ on monthly mean, representing about 5–29 % contribution to the calculated monthly mean $PM_{2.5}$ concentrations and contribution from the regional shipping dominated in the months from winter to summer (Fig. 11).

In the chemistry mode of TAPM, simplified chemical reactions for the secondary PM are included and the secondary particulate matter consists of organic carbon, reactive nitrogen and sulfate. While in summer more intensive photochemistry favors formation of precursors to the secondary PM, the air temperature and humidity controlled gas/particle partitioning of ammonium nitrate causes higher PM-nitrate concentrations during periods of the year with cold and wet weather. Many city-scale models do not involve chemistry and thus neglect formation of the secondary PM. Therefore, a sensitivity run was performed to investigate the role of the formation of secondary PM from local shipping on the city scale where only emissions of the primary PM were introduced, without emissions of the gas-phase pollutants from the local shipping. Modelled secondary PM concentrations from shipping were calculated as the difference between the base run and this sensitivity run. They were found relatively low, with maximum 0.0009 µg m$^{-3}$ in winter, and spread out since secondary PM is mainly formed far from the sources. The secondary PM tends to disperse and accumulate to the east part of Gothenburg due to the prevailing wind directions (Fig. A4 in the Appendix).

### 4 Calculation of exposure and health effect from ship emissions

The contribution of emission sources to population exposure depends on the relationship between population density and air pollution levels. The areas with relatively high exposure to $PM_{2.5}$ due to local and regional shipping are city ports and areas around, especially north of the Göta älv. Figure 12 presents the population weighted annual mean concentrations of $NO_2$, $PM_{2.5}$ and SOMO35 at each model grid for the base simulation and for contributions of the local plus regional shipping, as well as for contributions of the road traffic. The spatial patterns of $PM_{2.5}$ exposure from shipping are dominated by gradients in the concentration fields around the city ports to the north of Göta älv. $PM_{2.5}$ exposure from shipping is higher than exposure from road traffic in a larger city area since regional-shipping-related $PM_{2.5}$ exposure is evenly distributed over the city (Fig. A5 in





Appendix). The sum of PWC of $PM_{2.5}$ from the local plus regional shipping is 0.51 µg m-3 in the model domain, to which the regional shipping contributes with 82 %, comparing to 0.22 µg m-3 associated to road traffic (Table 2). The total exposure to $PM_{2.5}$ is dominated by particles transported to the city with the background air. The sum of PWC of $NO_2$ from regional and

local shipping was 1.65 ppb, similar to that from the road traffic (1.75 ppb), with gradients in the concentration fields north of the Göta älv. Because of the effect of local $O_3$ titration by the shipping emitted NO, the exposure to SOMO35 from shipping was negative along the Göta älv. However, SOMO35 exposure due to regional shipping was positive with sum of PWC 70.9 ppb×h in the model domain and showed relatively high level in areas with high population density.

The PWC for these pollutants were then used in the health impact calculations and results are presented as life years lost per year and loss of life expectancy (years of lifetime lost per person, YOLLs pers.[-1]) for $PM_{2.5}$ and as premature deaths for ozone and $NO_2$. The estimated loss of life expectancy (YOLLs pers.[-1]) due to $PM_{2.5}$ from local shipping was 0.003 while from the regional shipping it was 0.014. For comparison, impact of exposure to $PM_{2.5}$ from the road traffic was calculated to be 0.007 YOLLs pers.[-1] and to all $PM_{2.5}$ in the Base simulation to be 0.14 YOLLs pers.[-1] (Table 3). In all, shipping contributed with 12

% to the calculated health impacts from the total exposure to $PM_{2.5}$ in the city and the impact was more than 2 times larger than that of the local road traffic, the regional shipping being a larger risk for human health than the local shipping (> 80 %) in Gothenburg. The exposure to ozone related to shipping emissions reduced the acute mortality by 0.4 premature deaths per year due to the NO titration effect. This effect included additional 0.03 deaths attributed to ozone formed from the regional shipping emissions (Table 3). Exposure to $NO_2$ related to shipping emissions caused additional 2.6 premature deaths year[-1],

impact of the local shipping being similar to the regional one. This impact corresponded to 35 % of the impact of the $NO_2$ exposure in the Base simulation and was similar to the impact of the road traffic.

## 5 Assessment of uncertainties and comparison with other studies

Addressing uncertainties in human health risk assessment is a critical issue when evaluating the effects of contaminants on
public health due to the complex associations between environmental exposures and health. Uncertainties are introduced with the calculated pollutant concentrations, the grid resolution when assessing the population exposure, the general shape of concentration-response function and transferability problems of the function from region to region. Hammingh et al. (2012) presented an estimate of the uncertainty in the calculations of YOLLs, which may stem from the methodology used in the YOLL calculations and from the spatial resolution. To compare results of Jonsson et al. (2015) with results of this study, the
YOLLs pers.[-1] from $PM_{2.5}$ exposure calculated in ARP were multiplied by the life expectancy of population above the age of 30, i.e. 50 years, and divided by the population in the model domain. The health impacts of $PM_{2.5}$ were also calculated using the RAINS methodology directly on the calculated $PM_{2.5}$ exposures. Results of both methods are presented, giving very similar results.



The largest uncertainties are associated with the exposure response functions (ERF) as such. In this study impacts for the mean values of ERFs are presented, the 95 % confidence interval for these functions is given in the Methods chapter. The ERFs used here are those recommended in WHO (2013a), for $PM_{2.5}$ ERFs with higher values for spatial analyses of air pollution and mortality were found by project Escape for European cohorts (Beelen et al., 2014) as well as for mortalities in Los Angeles (Jerrett et al., 2005, 17 % per 10 µg m-3, 95 % confidence interval 5–30 %). These ERFs are of very similar value and those

of Beelen et al. (2014) were used as alternative functions for estimates of broader uncertainty limits by Barregård et al. (2019). There are two important issues regarding the uncertainties associated with the ERFs. First, the air pollution represents a complex mixture and individual gases and particles are often correlated. The impacts on mortality calculated for the different pollutants therefore cannot be simply summed up. Second, the ERFs assume that all particulate matter has the same impact. There is increasing evidence of different ERFs for some compounds, primarily elemental and organic carbon (WHO 2013b).


The most robust relation between the air pollution and effects on human health is for particulate matter (WHO 2013b). In Swedish cities, also in Gothenburg, the main contribution to concentrations of $PM_{2.5}$ comes from the background air (Segersson et al., 2017; Gustafsson et al. 2018). Accurate modelling of the total concentration of particulate matter is, however, very difficult as the processes affecting them are extremely complex and many of them not well quantified. These include natural

and anthropogenic emissions, formation of secondary particulate matter in complex photochemical processes as well as dry and wet deposition processes that need to be described on the whole range of relevant geographical and time scales. Many regional- and global-scale models tend to underestimate the simulated $PM_{2.5}$ concentrations, especially in summer, when formation of secondary PM is stronger due to the high photochemical activity and the impact of primary PM is lower due to the more intensive mixing and smaller anthropogenic emissions of primary PM in summer (Karl et al., 2019a). Also, the

modelled PM concentrations used as the boundary conditions in this study showed underestimates of $PM_{2.5}$ by 60 % and 17 %, on summer average and on annual average, respectively (Karl et al., 2019a). Two studies addressing impacts of shipping on air pollution in Gothenburg (Segersson et al., 2017; Repka et al., 2019) assessed the total concentration levels and contribution of shipping to these, none of them are, however, calculated for year 2012 assessed in this study. Segersson et al. (2017) shows annual mean background $PM_{2.5}$ concentrations for year 2011 of about 5 µg m-3 for Gothenburg, reaching

concentrations > 8 µg m-3 in polluted parts of the city. An annual mean concentration map presented in Repka et al. (2019) for year 2016 shows similar concentration levels with background concentrations of about 6 µg m-3 and maximum concentrations > 8 µg m-3. Both studies used $PM_{10}$ monitoring data at the urban background station 'Femman' to inversely derive the boundary conditions for $PM_{2.5}$. Jonson et al. (2015 and 2019) studied impacts of the Baltic Sea and North Sea shipping with the EMEP model for year 2010 and 2016 and found in both cases annual mean concentration levels at the Swedish West coast about 4–5

µg m-3. This concentration should correspond to the background levels of the city-scale simulations and year 2016. Jonson et al. (2019) also compared the modelled concentrations with background measurements from the station Råö, situated 20 km south of the city, and found a model underestimation of 0.7 ppb for the annual mean. The concentration levels of $PM_{2.5}$ found in this study were lower than in Segersson et al. (2017) and Repka et al. (2019), but they agree reasonably well with Jonson et

al., 2015 and 2019. Segersson et al. (2017) addressed health effects of $PM_{2.5}$, $PM_{10}$ and black carbon in three Swedish cities, among them Gothenburg using gaussian model SIMAIR. The population weighted exposure to $PM_{2.5}$ for Gothenburg was calculated to 6.5 µg m$^{-3}$, which was associated with c.a. 150–290 premature deaths from exposure to $PM_{2.5}$. The lower premature death number in Segersson et al. (2017) comes from calculations using the same ERF as in this study while the higher number uses the ERF presented by Jerrett et al. (2005) for $PM_{2.5}$ from the city sources. The values can be compared to the population weighted exposure to $PM_{2.5}$ of 4.1 µg m$^{-3}$, associated to c.a. 140 premature deaths found in this study. Jonson

et al. (2015) calculated impact of shipping emissions in the Baltic Sea and the North Sea using the EMEP model and a map presenting geographical distribution of life expectancy loss shows approximately 0.2 YOLLs pers.$^{-1}$ around the West coast of Sweden. This agrees reasonably well with our estimate of 0.18 YOLLs pers. $^{-1}$ in Gothenburg.

It is important to bear the uncertainties in total concentrations of PM and other air pollutants in mind when assessing the

relative contribution of shipping to the overall impact of air pollution. Assessments of impacts of selected anthropogenic sources are, however, associated with smaller uncertainties compared to the impact of the total concentrations as some large uncertainties, e.g. those regarding the natural and agriculture sources, cancel out. The study of Segersson et al. (2017) found contribution of shipping to the population weighted annual $PM_{2.5}$ concentration to be 0.04 µg m$^{-3}$ and contribution of the road traffic exhaust emissions to be 0.27 µg m$^{-3}$ which can be compared to 0.09 µg m$^{-3}$ from shipping and 0.22 µg m$^{-3}$ from road

traffic found in this study, however, bearing in mind that the studies assessed two different years.

## 6 Conclusions

The impact of local and regional ship emissions on in the city of Gothenburg was investigated by a multi-model system for the year 2012. The model evaluation against monitoring data demonstrated fairly good agreement in meteorological parameters and acceptable estimation of hourly air pollutant concentrations.


The city-scale model simulations with and without local and regional shipping in the emission inventory revealed that impacts from shipping on air quality in Gothenburg were substantial. The calculated contribution from local shipping to $NO_2$ was 0.5 ppb on annual average, representing 14 % of calculated annual mean $NO_2$ concentration. Including contribution from regional shipping in the North Sea and the Baltic Sea, the total shipping contribution reached 1.5 ppb representing 41 % of calculated

$NO_2$ concentrations. The contribution from regional and local shipping was higher than that from road traffic around the area of the city ports. In an analysis of exposure from different sources using population weighted concentrations, the contribution of regional and local shipping was similar to that of the road traffic in the city.



The model results of ozone concentrations have shown that titration by NO dominated the overall impact of local shipping on
ozone concentration levels in Gothenburg. The maximum impact from local $NO_x$ and NMVOC emissions on summer seasonal mean ozone concentration was calculated to -4 ppb. The negative effect of solely $NO_x$ emissions from local shipping on the ozone concentrations was up to -6 ppb, net negative even in the summer, when photochemical activity and potential for ozone formation are high. The emissions of NMVOC from local shipping as such increased the ozone formation in the city with the highest contribution of 2 ppb as seasonal summer mean. In terms of urban air quality control, reduction in anthropogenic
NMVOC could result in a significantly greater decrease in $O_3$ relative to the same reduction in $NO_x$ (Karl, 2019b).

The simulated emissions from local and regional shipping contributed 0.5 µg m$^{-3}$ on model/domain average and at highest 1.1 µg m$^{-3}$ to the annual mean concentration of $PM_{2.5}$. Regional shipping is a larger contributor than local shipping to local $PM_{2.5}$ concentrations, corresponding to 11 % of the local $PM_{2.5}$ concentrations on average. Also its contribution to the PWC was
higher, contributing with 0.4 µg m$^{-3}$ (10 % of the total PWC for $PM_{2.5}$). Contribution from the local shipping was 0.1 µg m$^{-3}$ (2 % of the total).

The calculated health impacts have shown the most serious effects from shipping in Gothenburg to be associated with exposure to $PM_{2.5}$. Local and regional shipping together reduce life expectancy by 0.015 years per person, of which more than 80 % are
associated with the regional shipping in the North and the Baltic Sea. The shipping impact is more than twice as high as the modelled impact of $PM_{2.5}$ associated with the local road traffic. Impacts from exposure to $NO_2$ and ozone were calculated in terms of premature deaths per year and 2.6 additional cases year$^{-1}$ were calculated for exposure to $NO_2$, regional and local shipping contributing with 59 % and 41 % respectively. Impacts from exposure to ozone were of opposite magnitude. The decrease of ozone due to the NO titration reduced the calculated mortalities by 0.4 cases year$^{-1}$. The impact of the exposure to
$PM_{2.5}$ from shipping calculated as premature deaths was 18 cases year$^{-1}$. The implementation of the more stringent SECA regulations on FSC in year 2015 is not likely to have changed impacts from $NO_2$ and ozone. According to the study of Jonson et al. (2019), approximately 35 % reduction of the impact from the regional shipping contribution to $PM_{2.5}$ could be expected around Gothenburg while a much smaller change can be expected in emission from the local shipping since hotelling and inland shipping already use a fuel with 0.1 % FSC in the model. This would mean similar reductions of the impacts related to
$PM_{2.5}$ in the city of Gothenburg. Impact of the global cap of 0.5 % for FSC which entered into force the 1$^{st}$ of January 2020 will not have any significant impact on further reduction of shipping-related air pollution in Gothenburg comparing to situation after 2015. Global study of Sofiev at al. (2018) shows that around the Swedish West coast decrease of $PM_{2.5}$ due to the global cap would be below 1%. The more serious health effects induced by regional shipping indicate that close cooperation across governance levels is required to effectively reduce the air pollution in the city.


Impacts of the local shipping emissions on air quality and human health are further discussed in part II paper (Ramacher et al., 2019, the Part II paper), presenting study of several future shipping scenarios for year 2040 adopting changes in shipping emissions due to changes in ship traffic volumes, legislation on emissions of air pollutants at sea, on energy effectivization as well as introduction of shore side electricity in shipping.


This article is part of the special issue "Shipping and the Environment 2017"

**Code availability**

The TAPM model is a commercial software available at CSIRO, Australia ([www.csiro.au](www.csiro.au)). STEAM model is intellectual property of the Finnish Meteorological Institute and is not publicly available. ARP is commercial software available from
arirabl.com.

**Data availability**

The model output data are available upon request from the corresponding authors

**Author contribution**

LT, MOPR, JM and VM designed the model simulations. LJ and JPJ calculated ship emissions with the STEAM model and
contributed with text about the shipping emissions, LT prepared ship emission files for the model simulations. LT, MG and JM prepared emission data from other sources. MK and AA prepared data from the regional-scale simulation used for the boundary conditions and MK contributed with text about these simulations. LT and MOPR prepared the model set-up and other input data, performed the model simulations and evaluated the model results. LT calculated exposures and JM and KY calculated the health impacts. LT and JM wrote the major part of the text with assistance from MOPR and VM.

**Competing interests**

JM is associated editor of the special issue Shipping and Environment.

**Acknowledgements**

Hulda Winnes and Stefan Åström, IVL, are acknowledged for valuable comments to the manuscript.





**Financial support**

This work has been conducted within the BONUS SHEBA (Sustainable Shipping and Environment of the Baltic Sea region) research project under Call 2014-41. BONUS (Art 185), funded jointly by the EU, Swedish Environmental Protection Agency, Academy of Finland and by the German Federal Ministry of Education and Research under Grant Number 03F0720A and within project platform CSHIPP, subsidy contract #C006 of Interreg Baltic Sea Region.

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

**Figure 1: Annual local ship emissions of (a) NO$_x$ and (b) PM$_{10}$ (equal to PM$_{2.5}$) for small vessels with stack height below 36 m (assumed 15 m) and (c) NO$_x$ and (d) PM$_{10}$ large vessels with high stacks height above 36 m (assumed 36 m) in the Gothenburg area. Base map credits: © OpenStreetMap contributors (openstreetmap.org). Distributed under a Creative Commons BY-SA License.**






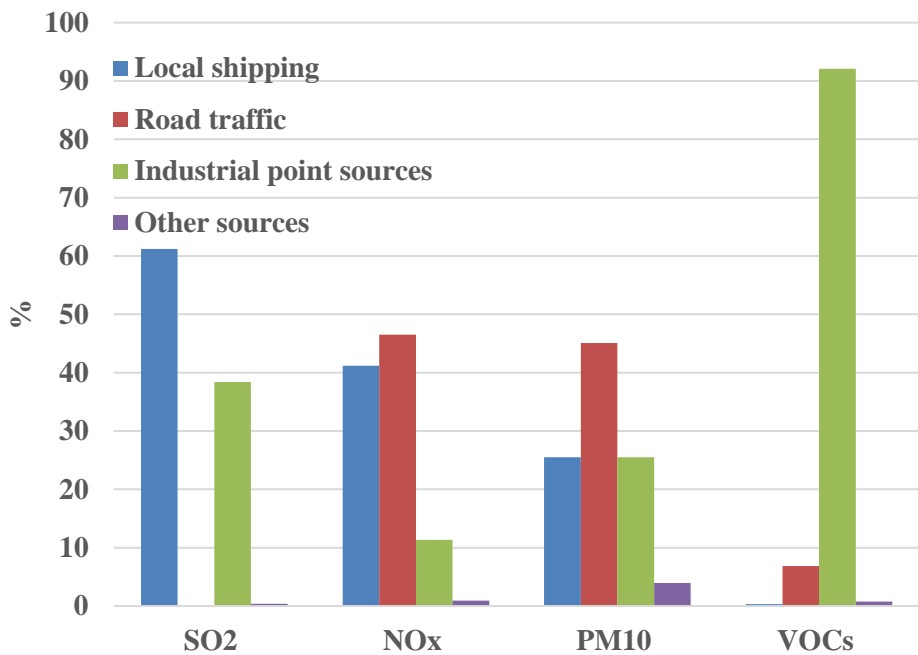


**Figure 2: Proportion of different source categories to the local emission inventory for the city-scale model domain in year 2012. The total emissions were 502 ton year⁻¹ for SO₂, 5072 ton year⁻¹ for NOₓ, 357 ton year⁻¹ for PM₁₀ and 7457 ton year⁻¹ for VOCs.**

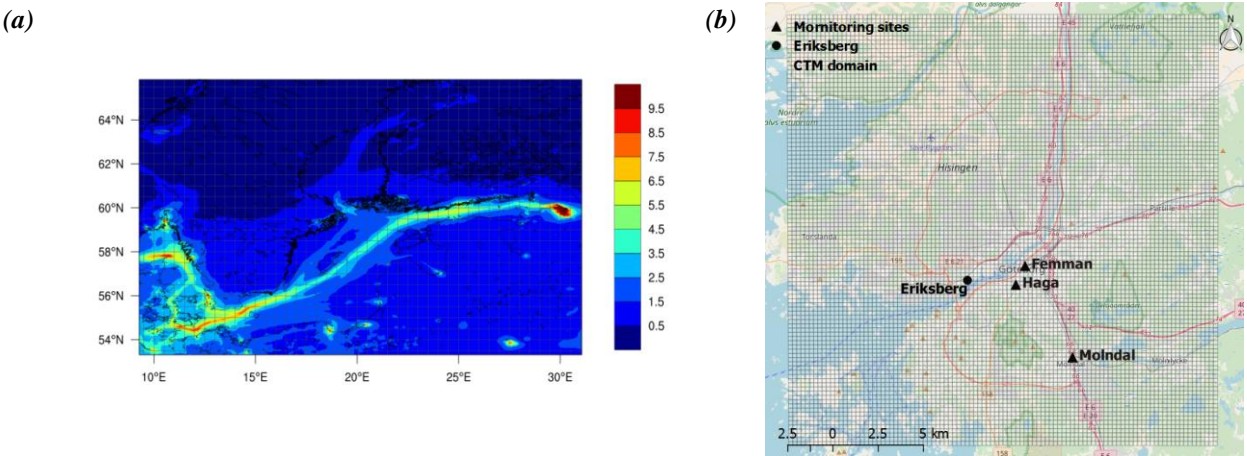

**Figure 3: Summer mean (JJA) NO₂ atmospheric concentrations [ppbV] from the regional-scale CMAQ simulation (4 × 4 km²) for year 2012 (a) used for the boundary conditions in TAPM, and (b) the TAPM domain (250 x 250 m²) with the three monitoring sites Femman, Haga and Mölndal used for validation of the urban-scale model simulations for Gothenburg. Eriksberg is the selected site which is close to the harbour and has high population density. Base map credits: © OpenStreetMap contributors (openstreetmap.org). Distributed under a Creative Commons BY-SA License.**





*(a)*   Wind rose plots of measurements at Femman station

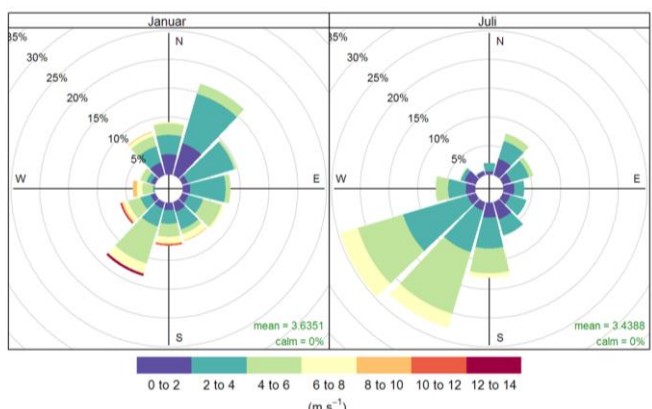

*(b)*

Difference wind rose plots at Femman station

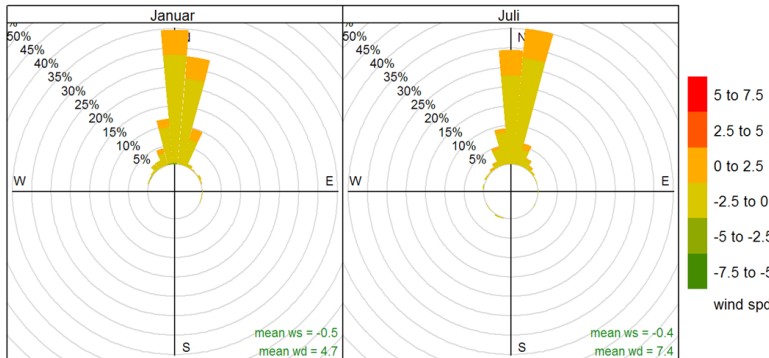


**Figure 4: Comparison between measured and modelled winds at Femman station: (a) the observed wind rose in January and July; (b) the differences between observed and simulated wind rose, showing the BIAS in wind speed based on the difference simulated wind speed – measured wind speed. E.g. A positive BIAS from 0 to 2.5 m s$^{-1}$ in wind direction N has a frequency of almost 30 %.**

 

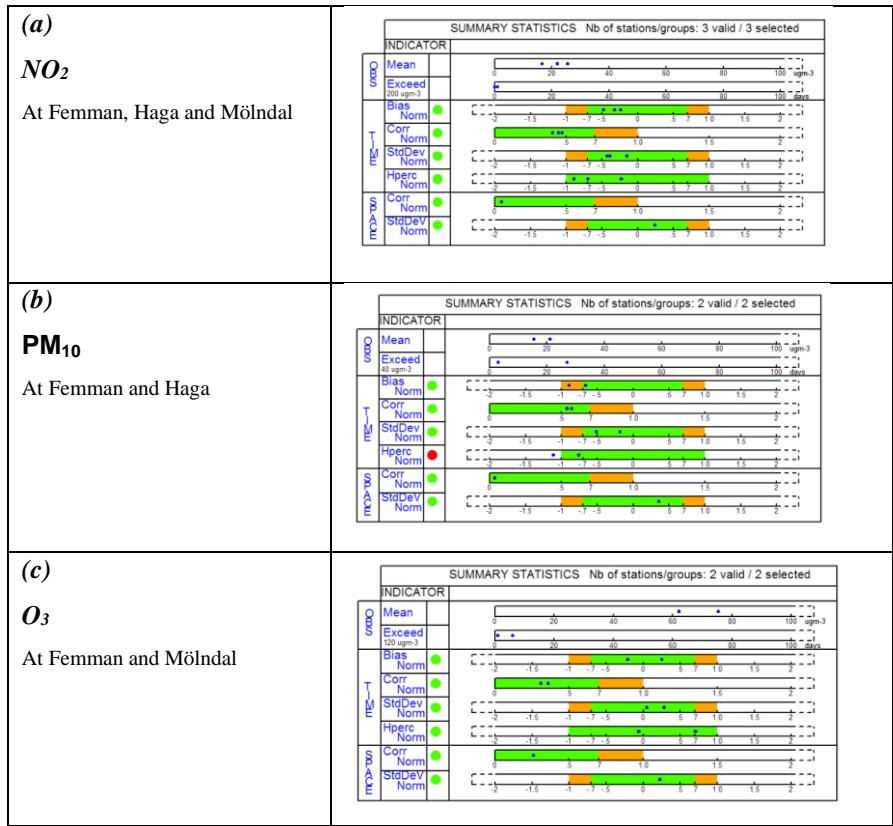


**Figure 5: Summary statistics of model performance for the annual mean values of NO₂ (hourly values), PM₁₀ (daily values) and O₃ (daily maximum of the 8-hour means), including days of exceedances. Stations (blue dots) within green bar: performance criteria satisfied, stations within orange bar: performance criteria satisfied, error dominated by the corresponding indicator. Green light: > 90 % of the stations fulfill the performance criteria. Red light: < 90 % of the stations fulfill the performance criteria. The indicator Hperc indicates the model capability to reproduce extreme events, represented by selected high percentile for modelled and observed values.**





*(a)*

*(b)*

*(c)*

*(d)*

**Figure 6: Simulated atmospheric concentrations of SO₂ (in ppb) for year 2012: (a) Annual mean concentrations in Base case simulation; (b) Annual mean contribution of the local shipping; (c) Annual mean contribution of the local and the regional shipping; and (d) Same as (c) with main ports along the Göta älv as well as Eriksberg. Base map credits: © OpenStreetMap contributors (openstreetmap.org). Distributed under a Creative Commons BY-SA License.**





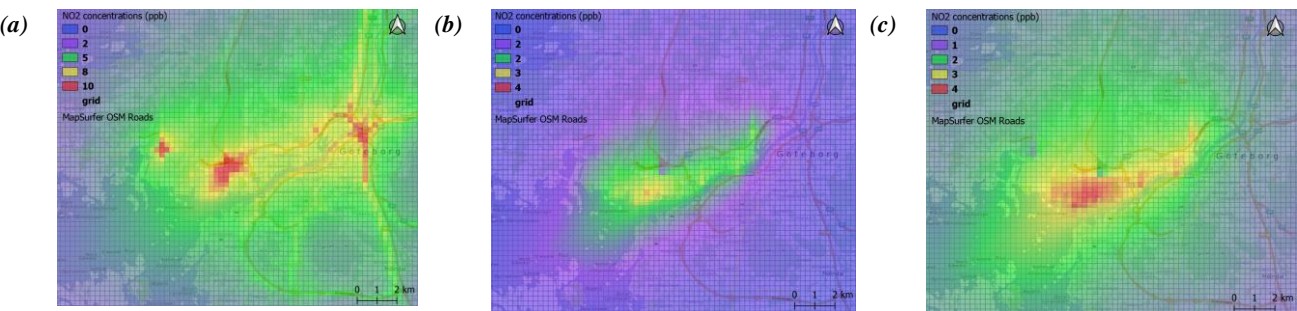

**Figure 7: Simulated atmospheric concentrations of NO₂ (in ppb) for year 2012: (a) Annual mean concentrations in Base case simulation; (b) Annual mean contribution of the local shipping; (c) Annual mean contribution of the local and the regional shipping. Base map credits: © OpenStreetMap contributors (openstreetmap.org). Distributed under a Creative Commons BY-SA License.**

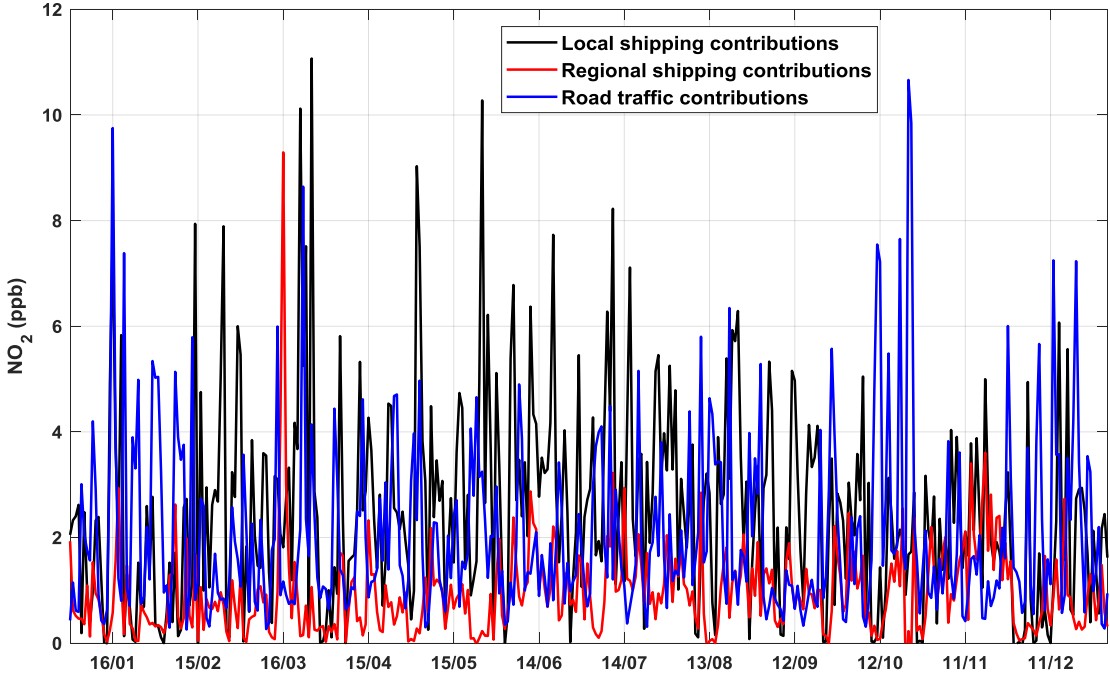

**Figure 8: Time series of simulated contributions of shipping and road traffic to NO₂ daily mean concentrations at Eriksberg, located at the north of the Göta älv for the year 2012. Modelled local shipping contributions (black line) deduced from the scenarios "Base" and "No local shipping", regional shipping contributions (red line) deduced from the scenario "No local shipping" and "No local and no regional shipping", and road traffic contributions (blue line) deduced from the scenario "Base" and "No road traffic" are presented.**





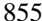

**Figure 9: Modelled summer mean (JJA) ozone concentrations (ppb) and contributions of local and regional shipping to the summer mean concentration in year 2012: (a) Modelled O₃ concentrations in the Base model simulation; (b) Modelled contribution of emissions from local shipping; (c) Modelled contribution of emissions from local and regional shipping; (d) Modelled contributions of NMVOC emissions from local shipping. Base map credits: © OpenStreetMap contributors (openstreetmap.org). Distributed under a Creative Commons BY-SA License.**





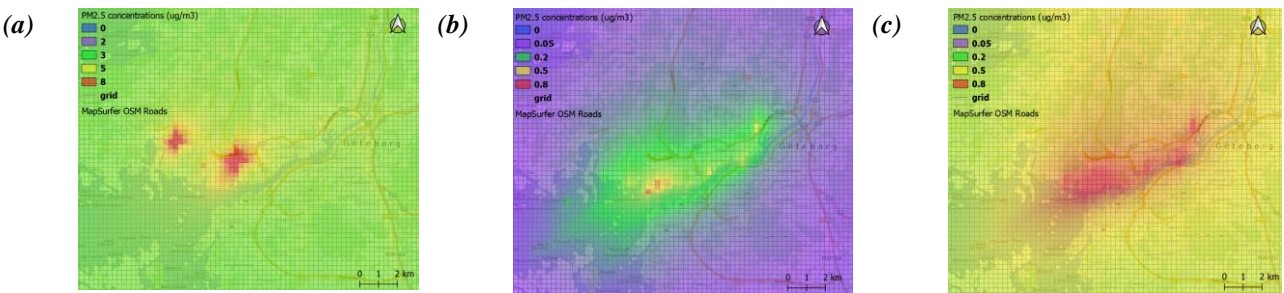

**Figure 10: Modelled annual mean PM$_{2.5}$ concentrations (µg m$^{-3}$) and contributions of shipping to the annual mean concentrations in year 2012: (a) Modelled annual mean concentrations in Base model simulation; (b) Modelled annual mean contribution of local shipping; (c) Modelled annual mean contributions of local and regional shipping. Base map credits: © OpenStreetMap contributors (openstreetmap.org). Distributed under a Creative Commons BY-SA License.**

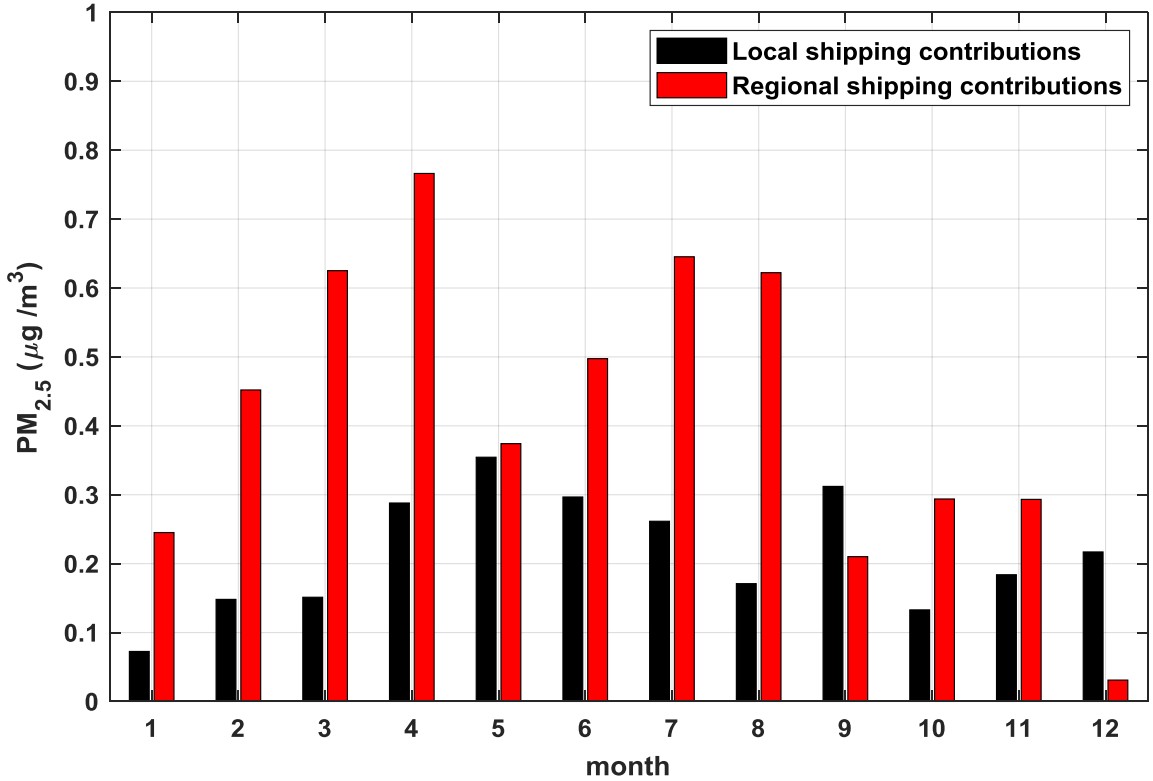

**Figure 11: Modelled monthly mean contributions to PM$_{2.5}$ concentrations (µg m$^{-3}$) at Eriksberg, Göta älv for the year 2012. Modelled local shipping contributions (black bar) are deduced from the differences between the scenarios "Base" and "No local shipping" and modelled regional shipping contributions (red bar) are deduced from the difference between the scenario "No local shipping" and "No local and no regional shipping".**





**Figure 12: The population weighted annual mean concentrations for NO₂ (ppb × capita), PM₂.₅ (µg m⁻³ × capita) and SOMO35 (ppb × h × capita): (a) NO₂ in the Base simulation; (b) PM₂.₅ in the Base simulation; (c) SOMO35 in the Base simulation; (d) NO₂ from local and regional shipping; (e) PM₂.₅ from local and regional shipping; (f) SOMO35 from local and regional shipping; (g) NO₂ from road traffic; (h) PM₂.₅ from road traffic and (i) SOMO35 from road traffic. Base map credits: © OpenStreetMap contributors (openstreetmap.org). Distributed under a Creative Commons BY-SA License.**



880

**Table 1: City-scale model setup.**

|  | Domain | Spatial resolutions | Model / Database |
|---|---|---|---|
| **Meteorology** | 30 km × 30 km | 500 m | ECMWF ERA5 0.3˚ × 0.3˚, 21 layers |
| **Background concentrations** | 160 km × 96 km | 4 km × 4 km | CMAQ |
| **Local shipping emissions** | 30 km × 30 km | 250 m × 250 m | STEAM2 |
| **Local traffic emissions** | 30 km × 30 km | meters (line sources) | Miljöförvaltningen and HBEFA v. 3.2 |
| **Local industrial, machines, wood burning and aviation *etc*.** | 30 km × 30 km | 1 km × 1 km | SMED |

**Table 2: Population weighted annual mean concentrations of $NO_2$, $PM_{2.5}$ and SOMO35 associated to all sources, road traffic and**
885 **local, regional shipping in city of Gothenburg for year 2012.**

| Sources | $NO_2$ (ppb) | $PM_{2.5}$ (µg m⁻³) | SOMO35 (ppb × h) |
|---|---|---|---|
| **All sources** | 4.70 | 4.12 | 19698.28 |
| **Road traffic** | 1.75 | 0.22 | 11.79 |
| **Local and regional shipping** | 1.65 | 0.51 | -1115.27 |
| *Local shipping* | 0.68 | *0.09* | *-1186.11* |
| *Regional shipping* | 0.97 | *0.42* | *70.85* |

**Table 3: Health impacts calculated for $O_3$, $NO_2$ and $PM_{2.5}$ contributions of the local and regional shipping and the local road traffic**
**to air pollution in the city of Gothenburg as well as of the total exposure to these pollutants in the city. The health impacts calculated**
890 **with the ARP model and with the RAINS methodology are presented.**

| Pollutant | Impact | Unit | Local shipping | NSBS regional shipping | All shipping | Local road traffic | Total exposure |
|---|---|---|---|---|---|---|---|
| $O_3$ | Acute Mortality (All ages) | Premature deaths | -0.5 | 0.03 | -0.4 | 0.005 | 7.6 |
| $NO_2$ | Acute Mortality (All ages) | Premature deaths | 1.06 | 1.52 | 2.59 | 2.73 | 7.35 |
| $PM_{2.5}$ | Chronic Mortality (All ages) | Life years lost | 31 | 143 | 174 | 74 | 1393 |
|  | Chronic mortality (All ages, ARP) | YOLLs pers. ⁻¹ | 0.003 | 0.013 | 0.015 | 0.007 | 0.123 |
|  | Chronic Mortality (Age 30+, RAINS) | YOLLs pers. ⁻¹ | 0.003 | 0.014 | 0.018 | 0.008 | 0.141 |
|  | Chronic mortality relative to that from the total exposure | - | 2 % | 10 % | 12 % | 5 % |  |