# Peer review of "The impact of ship emissions on air quality and human health in the Gothenburg area - Part I: 2012 emissions"

_Atmospheric Chemistry and Physics, 2020_

## Referee Comment (RC1) · Anonymous Referee #1 · 23 Mar 2020

The study investigates the health impacts due to exposure to air pollution due to local and regional shipping activities in the Gothenburg area in the year 2012, as well as the contributions of shipping to the overall pollutant levels over the area, using a fine resolution city scale air pollution model and a health impact assessment model. The paper is easy to follow, with clear presentation of the methodology and discussion of results. I favor the publication of the paper in ACP given my comments below are addressed.

Line 35-36: Is this the summer mean or annual mean, please rephrase the sentence.

Please provide relative contributions (%) along with absolute contributions

[Figure]

Line 157: "exposure-response function"

Line 296: What is spatial resolution of the SMED database that is the source of these "other" emissions and how are they regraded into 1x1 km resolution? In addition, how are there emissions regridded to the TAPM resolution of 250 m?

Section 2.2.2: Does road traffic include resuspension so that it is the largest PM10 source in the domain?

Section 2.4. Why no model evaluation for PM2.5? Section 3.2.4 includes some discussion on modelled vs measured PM2.5, why not include these in the model evaluation section? It is very important as PM2.5 is the main health impact pollutant and errors in PM2.5 simulations lead to underestimations in the health impacts.

Section 2.5: Are the age intervals taken into account? If not, please discuss potential shortcomings. How are the chronic vs acute impacts taken into account?

Section 3.2. Please provide with relative contributions along with absolute contributions throughout the text.

Section 5. How about the linearity of the ERF? There are studies clearly showing that assuming a linear relationship can lead to significant under or over estimation of health impacts depending on the concentration range. This should be discussed, I think.

Brandt et al., 2013 is not cited in the text.

Figure 8 could be made similar to figure 11, showing the monthly means as it is a bit crowded as it is now. In addition, both figures could use stacked bars instead. Finally, it would be great to create a similar figure where it shows the contributions from other pollutants as "others", that can be splitted into local and outside of Gothenburg and Sweden if possible, as in Im et al., ACP, 2019.

---

## Referee Comment (RC2) · Anonymous Referee #2 · 27 Mar 2020

This study combined regional and city-scale chemical transport models to investigate the effect of ship emissions on air quality and health in the port city of Gothenburg in 2012. The results of this study are interesting, however, there are a number of flaws preventing the acceptance of the manuscript at the current form. A substantial revision is needed to demonstrate the novelty of the study before it fits the quality of the journal.

Major comments: 1.About the model set-up I feel some information about the advection and diffusion needs to be described, so that people can understand how the air pollutants transport horizontally and vertically over the Eulerian grid. What is the model top pressure? How many layers are there in the TAPM? The authors mentioned that

only simple formation of secondary inorganic and organic aerosol exist in TAMP. How does that affect the simulation of PM2.5 and PM10? Any underestimation? I would imagine the ability of TAMP in reproducing particulate matters may not be as well as CMAQ. Please refer to my second concern, in which I strongly suggest a comparison be made between CMAQ and TAPM.

2. Model evaluation Figure 4. only shows the wind rose plots, and it is hard to tell whether the meteorological conditions perform well by the model. The authors mentioned that temperature, relative humidity, total solar radiation, wind speed, wind direction and precipitation shows high correlation and low bias. How low is the bias? Is it within certain criteria, i.e., temperature bias within half to one degree Celcius? The same applies to the evaluation of air quality variables. I feel it is very hard to read Fig. 5. The authors mainly show the annual mean comparison. How about daily scale? Any statistical metrics such as mean bias, mean normalized bias, etc. were calculated?

I think it is useful to construct either a time series comparison or scatter plot to give readers an overall impression how the model performs in terms of the daily scale, or even hourly scale, if possible.

How about the performance of the 4km * 4km CMAQ results? I believe it is interesting to do a comparison between the CMAQ results and the urban-scale model results, of course, together with the observations. Based upon this comparison, people can easily judge the usefulness of the ultra-fine scale city-level model. Currently, the city-scale model has a higher spatial resolution of 250-m, however, if the model performs worse than CMAQ, what is the major purpose of the ultra-fine resolution? The same applies to the meteorology. I don't feel the science was advanced by simply focusing on the city-scale model without detailed clarification of the advances of the model.

The authors evaluated the species of PM10, O3 and NO2, however, the health impact assessment is based on PM2.5, O3 and NO2. Why not evaluating PM2.5 directly? Line 493 mentioned that "In the chemistry mode of TAPM, simplified chemical reac-

tions for the secondary PM are included and the secondary particulate matter consists of organic carbon, reactive nitrogen and sulfate." I am also worried about the performance of PM2.5 in the TAPM since only simple secondary inorganic and organic aerosol scheme was applied. How about the aerosol modes? Is the model using bulk mode aerosol or sectional bin model in TAPM?

3. Fig.9 the effect of local emission on ozone The summer mean impact implies the NO titration effect. How about daily scale? The summer mean ozone is indeed quite low. Is there any day with slightly higher concentration, which may reveal different role of local shipping? It is not very persuasive by only using seasonal mean.

4. the section of 3.2 Impact of ship emissions on local air qualityMost of this section simply describes the figure by using domain average, which does not make too much sense and not too much useful. Some comparisons might be made with either other sources or other studies to reveal the advancement of this study. For instance, what do the contributions from the local and regional shipping emission tell us? Is it useful in future strategies in the control policy? Only simple descriptions greatly discount the value of the study.

Minor comments: 1. Line 48 Our study show "show" changed to "shows" 2. Line 49: emphasising changed to emphasizing 3. Line 157: exposureresponse Please add a space between exposure and response 4. Line 233: In this study, the meteorological component of TAPM was driven by the recently published ECMWF ERA5 synoptic Since the COSMO-CLM model has higher meteorological model and TAPM was driven by CMAQ 4km * 4km, why not using COSMO-CLM drives TAPM?

5. Line 235: five nested domains What are the five domains? It is better to show a figure of the five nested domains. The authors also need to clarify the spatial resolutions of the five domains. Fig. 3b: the domain should be inferred in Fig. 3a, so the readers can tell where the domain of the finer resolution is.

6. Line 307: "NOX" should be replaced by "NOx". 7. Line 512: "$\mu$g m-3" should be

replaced by "$\mu$g m-3", and the same applies to Line 523 and 549. 8. Line 488: "A3 in Appendix", but the Appendix only have S3, not A3. The same issue applies to "Fig. A4 in the Appendix" on line 503 and 511. 9. Fig 8, the x-axis label needs to be changed. For instance, either all using the mid-day of the month, i.e., 15/01, or something else to make it easy to follow. 10. Figure captions can be more succinct. A lot of repetitive words.

---

## Author Comment (AC1) · 17 Apr 2020

**Reply to Reviewer 1:**

The authors would like to thank to Reviewer 1 for a thorough review of the manuscript and for the constructive comments. Here are our responses:

1. *Line 35-36: Is this the summer mean or annual mean, please rephrase the sentence. Please provide relative contributions (%) along with absolute contributions*

Response:

It should be summer mean. The sentence has been rephrased as "The local shipping emissions of NOx led to a decrease of the summer mean O3 levels in the city by 0.5 ppb (~2%) in average."

2. *Line 157: "exposure-response function"*

Response:

 The typo has been corrected

3. *Line 296: What is spatial resolution of the SMED database that is the source of these "other" emissions and how are they regraded into 1x1 km resolution? In addition, how are there emissions regridded to the TAPM resolution of 250 m?*

Response:

The spatial resolution of the SMED database is 1x1 km. The SMED gridded emissions for the different source categories were applied directly as gridded sources in the model.

4. *Section 2.2.2: Does road traffic includes resuspension so that it is the largest $PM_{10}$ source in the domain?*

Response:

The road-traffic emissions include the wear particles, however, not the re-suspension. This fact has been added to the methods part.

5. *Section 2.4. Why no model evaluation for PM2.5? Section 3.2.4 includes some discussion on modelled vs measured PM2.5, why not include these in the model evaluation section? It is very important as PM2.5 is the main health impact pollutant and errors in PM2.5 simulations lead to underestimations in the health impacts.*

Response:

We would like to thank the reviewer for pointing out this omission. Model evaluation for $PM_{2.5}$ has been added to the text and Figure 5 extended with panel for summary statistics for $PM_{2.5}$. In addition, section on comparing measured versus modeled daily concentrations of $NO_2$, $O_3$, $PM_{10}$ and $PM_{2.5}$ has been included in the supplement in response to a comment of Reviewer 2.

[Figure]

**(c)**
**PM$_{2.5}$**
At Femman and Haga

6. *Section 2.5: Are the age intervals taken into account? If not, please discuss potential shortcomings. How are the chronic vs acute impacts taken into account?*

Response:

We would like to thank the reviewer for pointing out these deficiencies in description of the HIA methodology. Both aspect of the age intervals and form of the ERF for PM$_{2.5}$ have been added to Section 2.5:

ARP uses linear ERFs, recognizing the limited range of pollutant exposures in Europe. The YOLLs are calculated per year, applying the relative risk within national life tables. This is done through relation between life years lost per 100 000 population per unit PM$_{2.5}$ concentration and life expectancy of the population developed by Miller et al. (2003) based on analysis of life tables. The premature deaths are calculated using the total national mortality rate. This methodology is justified for European countries with health status and proportion of natural mortality of population corresponding to population studied in the epidemiological studies which brought forward the CRFs for all-cause mortalities. For regions with high concentration levels of PM$_{2.5}$ the HIA studies need to use different form of ERFs and for populations with different health status comparing to the US and Western Europe, cause-specific rather than all-cause mortalities need to be used.

7. Section 3.2. Please provide with relative contributions along with absolute contributions throughout the text.

Response:

The relative contributions were added for SO$_2$ concentrations (3.2.1), NO$_2$ (3.2.2.) and ozone (in Conclusions). For PM (3.2.4) the relative contributions were already presented.

8. *Section 5. How about the linearity of the ERF? There are studies clearly showing that assuming a linear relationship can lead to significant under or over estimation of health impacts depending on the concentration range. This should be discussed, I think.*

Response: The discussion on ERF model as well as on use of all-cause or cause-specific mortality ERFs was added to Section 5:

In ARP a linear form of ERFs is applied which is justified by a rather narrow interval of PM exposure levels in Europe. In terms of impact of the total exposure to PM$_{2.5}$ on natural mortality, the linear and log-linear form of the functions give similar results within the concentration range of 10–30 µg m$^{-3}$, the linear model giving slightly lower relative risks in this range and higher relative risks below and above (Ostro et. al., 2004). The PM$_{2.5}$ levels found in our study fall below 10 µg m$^{-3}$. For regions with high PM$_{2.5}$ levels different ERF models need to be applied and for HIA global studies or studies

in other regions bur Western Europe or North America also ERFs for cause-specific mortalities, rather than natural mortalities are usually used.

In terms of incremental effects, the impacts can differ substantially between the two models at different concentration levels. Sofiev et al. (2018) show difference in relative risks of cause-specific mortalities for different base concentrations, at 1 µg m$^{-3}$ the log-linear model gives higher incremental relative risks than the linear model while at 5 and 10 µg m$^{-3}$ levels the log-linear model gives lower incremental relative risks.

9. *Brandt et al., 2013 is not cited in the text.*

Response:

The reference has been deleted

10. *Figure 8 could be made similar to figure 11, showing the monthly means as it is a bit crowded as it is now. In addition, both figures could use stacked bars instead. Finally, it would be great to create a similar figure where it shows the contributions from other pollutants as "others", that can be split into local and outside of Gothenburg and Sweden if possible, as in Im et al., ACP, 2019.*

Response:

The Figure 8 as well as Figure 11 has been updated by showing the contributions to monthly mean NO$_2$ and PM$_{2.5}$ concentrations from local shipping, regional shipping, road traffic and others.

[Figure]

*Figure 8: Modelled monthly mean contributions of the local shipping, regional shipping, local road traffic and other anthropogenic emissions (including contribution from the boundary conditions) to the NO2 concentrations (ppb) at Eriksberg in year 2012.*

---

## Author Comment (AC2) · 17 Apr 2020

**Reply to Reviewer 2:**

We would like to thank the reviewer for a thorough review of the manuscript and for many good points and suggestions for improvement. We have MET these valuable comments mainly by including additional information to the manuscript and believe that it gained more clarity, especially in terms of the modelling methodology employed and its verification. In the following text, the comments are answered and the changes in the manuscript indicated. The Response includes some new citations. Those which are not included in the manuscript are included at the end of the Response.

*Major comments:*

1. *About the model set-up I feel some information about the advection and diffusion needs to be described, so that people can understand how the air pollutants transport horizontally and vertically over the Eulerian grid. What is the model top pressure? How many layers are there in the TAPM?*

   Response:
   We would like to thank the reviewer for pointing out need of better description of the model. We have extended this part with more details and added information on vertical layers to the place where the model domain is described:

   TAPM consists of a meteorological and an air pollution components. The meteorological component of TAPM is an incompressible, non-hydrostatic, primitive equation model with a terrain-following vertical sigma coordinate for 3-D simulations. The model solves the momentum equations for horizontal wind components, the incompressible continuity equation for vertical velocity, and scalar equations for potential virtual temperature and specific humidity of water vapour, cloud water/ice, rain water and snow. The turbulence terms in these equations have been determined by solving equations for turbulence kinetic energy and eddy dissipation rate, and then using these values to represent vertical fluxes by a gradient diffusion approach (Hurley, 2008 b). Using predicted meteorology and turbulence from the meteorological component, TAPM applies Eulerian grid module in its air pollution component which consists of nested grid-based solutions of the Eulerian concentration mean equations representing advection, diffusion, chemical reactions and emissions. Dry and wet deposition processes are also included. (l. 227-235)

   In TAPM, an Exner pressure function is integrated from mean sea level to the model top (10 Pa in this study) to determine the top boundary condition. The Exner pressure function is determined from the sum of the hydrostatic component and non-hydrostatic component (Hurley, 2008). The number of vertical grid levels was 30 in this study. Twenty of these layers are below approximately 2 km; the lowest layer extends to ca. 10 m above ground. (l. 248-251)

2. *The authors mentioned that only simple formation of secondary inorganic and organic aerosol exist in TAMP. How does that affect the simulation of PM2.5 and PM10? Any underestimation? I would imagine the ability of TAMP in reproducing particulate matters may not be as well as CMAQ. Please refer to my second concern, in which I strongly suggest a comparison be made between CMAQ and TAPM.*

Response:

The secondary aerosol formation in TAPM is heavily parameterized, however, captures the important features of the secondary particle formation, i.e. formation of sulphate and nitrate following the $SO_2$ and $NO_2$ oxidation, as well as formation of SOA as a fixed part of the degraded smog reactivity representing VOC species in the reaction scheme of TAPM (Hurley, 2008b). We have replaced the description of aerosol formation in TAPM in the PAPER with this wording.

On urban scale, formation of secondary PM is usually supressed as the radical pool is depleted by the primary emissions and many urban models do not consider the secondary PM at all. We recognise that this assumption is questionable for shipping emissions, which are often emitted into relatively clean air masses coming from the sea over the harbour area to the city, additionally, also chemistry involving sea-salt aerosol particles can be of importance in this case. We have investigated contribution of secondary PM to the total PM modelled with TAPM photochemistry scheme, Fig. S4 shows contributions of max. 2% of the PM related to the local shipping in Gothenburg in winter months and negligible contributions in summer. In an earlier study (Haeger-Eugensson et al., 2010) we have compared oxidation processes in a ship plume transported over Gothenburg area simulated with TAPM photochemistry scheme and with much more detailed scheme including explicit aerosol chemistry of the MOCCA model (Sander et al., 1996, Pszenny et al., 2004) and found that during the day time the 2 schemes gave similar results while in dark hours the NO2 oxidation was underestimated, mainly due to the missing night-time NO3 chemistry. The MOCCA scheme does not, however, involve any advanced SOA chemistry so the performance of the two schemes regarding the SOA formation was not investigated.

The main idea of our city-scale study utilizing the boundary conditions of the CMAQ model simulations is to assess the urban-scale features of the shipping emissions, including differentiation of the regional shipping and the local shipping contributions to air pollution in the city. The regional-scale secondary PM formation is captured by the CMAQ model and is transferred to the local scale through the boundary concentration fields. CMAQ includes both, secondary inorganic (SIA) and secondary organic aerosol (SOA) formation. SIA formation builds mainly upon the widely distributed ISORROPIA mechanism (Nenes et al., 1998) and considers sulphate, nitrate, ammonium and interactions with sea salt. SOA can be formed from biogenic precursors (isoprene, terpenes, sesquiterpenes) and/or through oxidation of anthropogenic VOCs. As most regional modeling systems do, CMAQ typically underestimates PM concentrations, in particular SOA, because of unknown oxidation pathways or underestimated emissions (e.g. Solazzo et al., 2012).

The PM components of CMAQ, as well as the gases and radical species are re-calculated into the compounds included in TAPM. We expect that TAPM underestimates the secondary PM formation as discussed above; however, we don't expect that this effect is large on the urban scale.

3. *Model evaluation Figure 4. only shows the wind rose plots, and it is hard to tell whether the meteorological conditions perform well by the model. The authors mentioned that temperature, relative humidity, total solar radiation, wind speed, wind direction and precipitation show high correlation and low bias. How low is the bias? Is it within certain criteria, i.e., temperature bias within half to one degree Celsius?*

Response:
We would like to thank the Reviewer for pointing out the need to carry out a better evaluation of modelled meteorological parameters. Thus, we added a section on comparing measured versus modeled meteorological parameters (as shown below) in the supplement. Additionally, we have added a reference in the manuscript section 3.1, which directs the reader to the supplement.

Moreover, we would like to point to a study by Tang et al. (2009), which performed an evaluation and comparison (with MM5) of meteorological parameters on the urban-scale in Gothenburg. The results of that study showed that "(1) TAPM performs better than MM5 in simulating near-surface air temperature and wind in urban area, (2) both models are able to reproduce nighttime vertical temperature gradient reasonably well, but underestimate daytime temperature gradient, and (3) the two models significantly underestimate the occurrences of low wind speed situation at night. These results indicate that the performance of TAPM in simulating meteorological features over the urban area is generally comparable to that of MM5. TAPM can be used with some confidence to describe the local-scale meteorology needed for air quality applications." (Tang et al. 2009). Moreover, we applied urban-scale meteorology, simulated with TAPM, successfully in other harbor city studies (Ramacher et al. 2019, Ramacher et al. 2020). Evaluations in these studies also showed good performance of meteorological fields derived with TAPM. Table S1 has been added to the manuscript supplement.

*Table S1: Evaluation of modelled versus measured hourly meteorological parameters*

| parameter | site | n | MB | NMB | RMSE | r | IOA |
|---|---|---|---|---|---|---|---|
| Temp | all sites | 34261 | -0.46 | -0.06 | 2.09 | 0.96 | 0.87 |
| Temp | Femman | 8003 | -1.14 | -0.12 | 2.15 | 0.97 | 0.85 |
| Temp | GbgA | 8784 | -0.53 | -0.06 | 2.09 | 0.97 | 0.87 |
| Temp | Landvetter | 8783 | -0.03 | 0.00 | 2.27 | 0.96 | 0.86 |
| Temp | VingaA | 8691 | -0.20 | -0.02 | 1.81 | 0.97 | 0.88 |
| ws | all sites | 34004 | -0.18 | -0.04 | 0.51 | 0.99 | 0.93 |
| ws | Femman | 7772 | -0.17 | -0.05 | 0.26 | 0.99 | 0.93 |
| ws | GbgA | 8780 | -0.26 | -0.09 | 0.76 | 0.93 | 0.80 |
| ws | Landvetter | 8779 | 0.06 | 0.01 | 0.11 | 1.00 | 0.97 |
| ws | VingaA | 8673 | -0.35 | -0.05 | 0.61 | 0.99 | 0.92 |
| wd | all sites | 34008 | 2.35 | 0.01 | 46.31 | 0.87 | 0.93 |
| wd | Femman | 7776 | 1.18 | 0.01 | 24.63 | 0.96 | 0.97 |
| wd | GbgA | 8780 | 2.40 | 0.01 | 66.18 | 0.76 | 0.85 |
| wd | Landvetter | 8779 | 5.02 | 0.03 | 46.14 | 0.87 | 0.94 |
| wd | VingaA | 8673 | 0.66 | 0.003 | 35.72 | 0.92 | 0.96 |
| rh | all sites | 25457 | 2.73 | 0.04 | 12.49 | 0.64 | 0.59 |
| rh | Femman | 8003 | 6.02 | 0.08 | 13.70 | 0.67 | 0.57 |
| rh | GbgA | 8781 | 1.30 | 0.02 | 13.12 | 0.64 | 0.59 |
| rh | VingaA | 8673 | 1.15 | 0.01 | 10.51 | 0.65 | 0.61 |
| rain | all sites | 24935 | 0.32 | 3.37 | 0.87 | 0.29 | -0.15 |
| rain | Femman | 7772 | 0.39 | 4.00 | 0.99 | 0.29 | -0.26 |
| rain | GbgA | 8551 | 0.37 | 3.13 | 0.97 | 0.30 | -0.11 |
| rain | VingaA | 8612 | 0.22 | 2.98 | 0.62 | 0.26 | -0.06 |
| tsr | Femman | 7941 | 21.48 | 0.18 | 125.95 | 0.82 | 0.77 |

4. *The same applies to the evaluation of air quality variables. I feel it is very hard to read Fig. 5. The authors mainly show the annual mean comparison. How about daily scale? Any statistical metrics such as mean bias, mean normalized bias, etc. were calculated? I think it is useful to construct either a time series comparison or scatter plot to give readers an overall impression how the model performs in terms of the daily scale, or even hourly scale, if possible.*

Response:
We would like to thank the Reviewer for pointing out the need to carry out a better evaluation of modelled concentrations. Thus, we added a section on comparing measured versus modeled daily concentrations of $NO_2$, $O_3$, $PM_{10}$ and $PM_{2.5}$ in the supplement together with description of the indicators presented in the table (Supplement section S1). This section contains a table of relevant statistical parameters (Table S2), as well as scatter plots of modeled versus measured daily concentrations for all stations and pollutants (Figure S1). Additionally, we added a reference in the manuscript section 3.1, which directs the reader to the supplement and enhanced the manuscript text by adding values for underestimations, which are the main drawback of the modeled results in terms of their use in health-effect calculations.

Nevertheless, we decided to keep the summary statistics as calculated with FAIRMODE DeltaTool in the manuscript, due to the focus and aim of the DeltaTool to evaluate air quality modeling results for policy applications.

**S2 Statistical indicators and model performance indicators**

In the statistical analysis of the model performance, the following statistical indicators are used: normalized mean bias (NMB), standard deviation (STD), root mean square error (RMSE), correlation coefficient (r), index of agreement (IOA) and the fraction of predictions within a factor of two of observations (FAC2). The overall bias captures the average deviations between the model and observed data and the NMB is given by:

$$NMB = \frac{\overline{M} - \overline{O}}{\overline{O}}$$

where $\overline{M}$ and $\overline{O}$ stand for the averaged model and observation results, respectively. The RMSE combines the magnitudes of the errors in predictions for various times into a single measure and is defined as

$$RMSE = \sqrt{\frac{1}{N} * \sum_{i=1}^{N} (M_i - O_i)^2}$$

where subscript $i$ indicates the time step and $N$ the number of observations. RMSE is a measure of accuracy, to compare prediction errors of different models for a particular data and not between datasets, as it is scale-dependent. The correlation coefficient (Pearson r) for the temporal correlation is defined as:

$$r = \frac{\sum_{i=1}^{n}(O_i - \bar{O}) \cdot (M_i - \bar{M})}{\sqrt{\sum_{i=1}^{n}(O_i - \bar{O})^2 \cdot \sum_{i=1}^{n}(M - \bar{M})^2}}$$

The index of agreement is defined as:

$$IOA = 1 - \frac{\sum_{i=1}^{N}(O_i - M_i)^2}{\sum_{i=1}^{N}(|M_i - \bar{M}| + |O_i - \bar{O}|)^2}$$

An IOA value close to 1 indicates agreement between modelled and observed data. The fraction of modelled values within a factor of two (FAC2) of the observed values are the fraction of model predictions that satisfy is defined as:

$$0.5 \ \leq \frac{M_i}{O_i} \leq 2.0 \tag{9}$$

For evaluation of modelled values in rural areas, the acceptance criteria is FAC2 ≥ 0.5, while in urban areas it is FAC2 ≥ 0.3.

*Table S2: Evaluation of modeled versus measured daily concentrations of NO2, O3, PM10 and PM2.5*

| Site | period | n | FAC2 | MB | MGE | NMB | NMGE | RMSE | r | COE | IOA |
|---|---|---|---|---|---|---|---|---|---|---|---|
| **$NO_2$** | | | | | | | | | | | |
| Femman | annual | 346 | 0.71 | -7.58 | 9.24 | -0.34 | 0.42 | 12.68 | 0.50 | -0.03 | 0.48 |
| Femman | summer | 92 | 0.96 | -0.16 | 4.02 | -0.01 | 0.27 | 5.14 | 0.65 | 0.22 | 0.61 |
| Femman | winter | 90 | 0.43 | -15.35 | 15.88 | -0.53 | 0.55 | 19.27 | 0.46 | -0.46 | 0.27 |
| Haga | annual | 366 | 0.58 | -11.93 | 12.62 | -0.47 | 0.50 | 16.44 | 0.59 | -0.18 | 0.41 |
| Haga | summer | 92 | 0.76 | -8.06 | 8.15 | -0.39 | 0.39 | 9.93 | 0.76 | -0.12 | 0.44 |
| Haga | winter | 91 | 0.40 | -18.59 | 18.78 | -0.58 | 0.59 | 23.63 | 0.62 | -0.32 | 0.34 |
| Molndal | annual | 338 | 0.68 | -5.72 | 8.41 | -0.34 | 0.50 | 14.39 | 0.37 | 0.14 | 0.57 |
| Molndal | summer | 88 | 0.73 | 2.38 | 4.01 | 0.25 | 0.42 | 5.10 | 0.53 | 0.02 | 0.51 |
| Molndal | winter | 74 | 0.38 | -17.44 | 18.23 | -0.64 | 0.67 | 26.47 | 0.54 | -0.06 | 0.47 |
| **$O_3$** | | | | | | | | | | | |
| Femman | annual | 326 | 0.92 | -4.95 | 14.82 | -0.08 | 0.25 | 18.40 | 0.66 | 0.22 | 0.61 |
| Femman | summer | 92 | 0.95 | -12.26 | 16.44 | -0.19 | 0.26 | 19.89 | 0.53 | -0.23 | 0.38 |
| Femman | winter | 52 | 0.87 | 3.50 | 11.25 | 0.09 | 0.28 | 13.79 | 0.76 | 0.34 | 0.67 |
| Molndal | annual | 338 | 0.91 | 9.58 | 15.27 | 0.20 | 0.32 | 19.51 | 0.55 | -0.17 | 0.42 |
| Molndal | summer | 88 | 0.98 | 0.97 | 10.88 | 0.02 | 0.20 | 13.30 | 0.53 | -0.18 | 0.41 |
| Molndal | winter | 74 | 0.78 | 11.66 | 17.14 | 0.32 | 0.47 | 21.31 | 0.33 | -0.44 | 0.28 |
| **$PM_{10}$** | | | | | | | | | | | |
| Femman | annual | 324 | 0.56 | -6.80 | 7.96 | -0.43 | 0.51 | 10.39 | 0.24 | -0.45 | 0.28 |
| Femman | summer | 91 | 0.52 | -7.73 | 7.91 | -0.53 | 0.54 | 9.71 | 0.17 | -0.84 | 0.08 |
| Femman | winter | 59 | 0.63 | -4.25 | 7.25 | -0.27 | 0.45 | 9.13 | 0.28 | -0.12 | 0.44 |
| Haga | annual | 343 | 0.42 | -12.15 | 12.92 | -0.58 | 0.62 | 17.55 | 0.10 | -0.43 | 0.28 |
| Haga | summer | 79 | 0.23 | -16.36 | 16.40 | -0.72 | 0.72 | 21.40 | 0.17 | -0.69 | 0.15 |
| Haga | winter | 81 | 0.65 | -6.07 | 7.74 | -0.36 | 0.46 | 11.37 | 0.25 | -0.08 | 0.46 |
| **$PM_{2.5}$** | | | | | | | | | | | |
| Haga | annual | 343 | 0.42 | -3.31 | 4.09 | -0.44 | 0.54 | 4.96 | 0.59 | -0.59 | 0.21 |
| Haga | summer | 79 | 0.24 | -4.63 | 4.63 | -0.63 | 0.63 | 5.03 | 0.47 | -1.54 | -0.21 |
| Haga | winter | 81 | 0.53 | -3.14 | 4.26 | -0.36 | 0.49 | 5.29 | 0.50 | -0.41 | 0.30 |

(a)

Daily NO$_2$ [ppb]

[Figure]

(b)

Daily O$_3$ [ppb]

[Figure]

(c)

Daily PM$_{10}$ [$\mu$g m$^{-3}$]

[Figure]

(d)

[Figure]

*Figure S1: Scatter plots of measured versus observed daily (a) NO2, (b) O3, (c) PM10 and (d) PM2.5 concentrations.*

5. *How about the performance of the 4km * 4km CMAQ results? I believe it is interesting to do a comparison between the CMAQ results and the urban-scale model results, of course, together with the observations. Based upon this comparison, people can easily judge the usefulness of the ultra-fine scale city-level model. Currently, the city scale model has a higher spatial resolution of 250-m, however, if the model performs worse than CMAQ, what is the major purpose of the ultra-fine resolution? The same applies to the meteorology. I don't feel the science was advanced by simply focusing on the city-scale model without detailed clarification of the advances of the model.*

Response:
We would like to thank the Reviewer for pointing out the need to clarify the advantages of applying a city-scale model for the purpose of this study. As described in response to question 2, the main idea of our city-scale study is to utilize the boundary conditions of the CMAQ model simulations in a city-scale air quality model to assess the urban-scale features of the shipping emissions. In general, regional air quality models can give a reliable representation of concentrations in the urban background, but due to their limitation in resolving the near-field dispersion of emission sources and photochemistry at the sub-kilometre scale, around industrial stacks and on the neighbourhood level, they cannot provide the information needed by urban policymakers for population exposure mapping, city planning and the assessment of abatement measures. City-scale air quality models overcome the limitation inherent in regional-scale models by taking into account details of the urban topography, wind flow field characteristics, land use information and the geometry of local pollution sources. Thus, it is necessary to move beyond a resolution of e.g. 4km x 4km (resolution of CMAQ simulations used for the regional background). The city-scale air quality model TAPM was successfully applied to investigate urban air quality and scenarios in coastal urban areas all over the world (e.g. Matthias et al., 2018, Ramacher et al., 2020, Gallego et al., 2016, Fridell et al., 2014). Especially the meteorological module has proven to be capable of reproducing measured parameters such as temperature, wind speed and wind direction, because of its capability to capture meteorological effects sea-land circulations and complex terrain. This tackles also the minor comment on the use of COSMO-CLM instead of TAPM. COSMO-CLM does not take into account such effects on the urban-scale and thus, we decided to simulate and apply meteorological fields with TAPM. Previous studies (e.g. Tang et al. 2009 and Ramacher et

al. 2018) prove good meteorological simulation capabilities for the urban-scale, based on synoptic reanalysis, which we can confirm with the results of the presented study on Gothenburg (see also response to question 3).

When it comes to a possible comparison with other models on the regional scale, such as CMAQ, we would like to refer to a study by Karl et al. 2019, who compared the newly developed urban-scale CTM EPISODE-CityChem with TAPM and CMAQ. Karl et al. 2019 have carried out a full-year run with the TAPM air quality model to compare it with the urban-scale CTM EPISODE-CityChem. In this study, the TAPM run has been performed with the same horizontal resolution (1 km) as the EPISODE-CityChem run, identical emissions, but 2-D boundary concentrations instead of 3-D boundary conditions from CMAQ. CMAQ was not further included in the evaluation published in the manuscript because CMAQ cannot give realistic concentrations at the traffic sites and the industrial sites. CMAQ is a regional CTM system which does not handle local scale dispersion, i.e. a traffic site and a background site located within the same 4 x 4 km2 grid cell would have the same concentration values. If the traffic stations and industrial stations were included, it would be obvious that CMAQ fails to reproduce concentrations at urban stations that are impacted by the local pollution. A realistic representation of local emissions is complicated by their high the spatial and temporal variability in the urban area. Urban-scale CTM such as EPISODE-CityChem and TAPM use the local scale emissions to compute the pollutant concentrations in the urban background areas, which are in turn affected by the highly resolved emissions. Therefore, urban scale models are much more sensitive to an incorrect representation of the local emissions than a regional scale model with coarser resolution.

Finally, we decided not to take into account a comparison of concentrations simulated with TAPM and concentrations simulated with CMAQ, also because the focus of this study is not the comparison of regional and city-scale CTM performances but much more the local effects, trends and challenges that arise for the urban population and policy.

6. *The authors evaluated the species of $PM_{10}$, $O_3$ and $NO_2$, however, the health impact assessment is based on $PM_{2.5}$, $O_3$ and $NO_2$. Why not evaluating $PM_{2.5}$ directly? Line 493 mentioned that "In the chemistry mode of TAPM, simplified chemical reactions for the secondary PM are included and the secondary particulate matter consists of organic carbon, reactive nitrogen and sulfate." I am also worried about the performance of $PM_{2.5}$ in the TAPM since only simple secondary inorganic and organic aerosol scheme was applied. How about the aerosol modes? Is the model using bulk mode aerosol or sectional bin model in TAPM?*

Response:

Model evaluation for $PM_{2.5}$ has been added to the text and Figure 5 extended with panel for summary statistics for $PM_{2.5}$. Additionally, we added a section on detailed concentration evaluation in the supplement (see response to comment 4).

The secondary aerosol formation in TAPM is already discussed above at Discussion point 2. We have concluded that the regional-scale secondary aerosol formation is covered in the boundary concentrations calculated by CMAQ and that we don't expect that effect of simplified secondary aerosol formation is large on the urban scale. Regarding the aerosol scheme, TAPM is using refined bulk mode, having separate scheme for Fine PM corresponding to $PM_{2.5}$ and Ambient PM corresponding to $PM_{10}$, which both include secondary PM. Two additional modes for particles in size fractions 10-20 and 20-30 μm not

involving secondary aerosol formation are included, these have not been used in our simulations.

| (c)
**PM₂.₅**
At Femman and Haga |
[Figure]
 |
|---|---|

7. *Fig.9 the effect of local emission on ozone The summer mean impact implies the NO titration effect. How about daily scale? The summer mean ozone is indeed quite low. Is there any day with slightly higher concentration, which may reveal different role of local shipping? It is not very persuasive by only using seasonal mean.*

Response:

We would like to thank the reviewer for a very good suggestion to show the ozone formation due to the different sources on daily scale. We have added Fig. S5 in Supplement showing the modelled daily mean ozone concentrations contributed from local shipping, regional shipping and VOC emissions from local shipping at Eriksberg under summer and added the following text to the paper:

Further details of impact of the shipping emissions on ozone formation are illustrated in Figure S5 in the Supplement, showing summer ozone formation from regional and local shipping as well as from the local shipping VOC emissions at Eriksberg. At this location the local shipping emissions lead almost always to ozone depletion. On contrary, VOC emissions from local shipping cause the increase of ozone concentrations, confirming that the location is in a VOC-limited photochemical regime. The regional shipping tends to increase the local ozone concentrations in most of days (78 days under June–August). Inspecting details of the diurnal variation of ozone contributions (Figure S5b-d), one can see that during the rare occasions without ozone depletion by the local shipping, there is a small ozone formation from the local shipping emissions and no ozone formation from the local shipping VOC emission, indicating presence of NOx-limited regime (Fig. S5b), while during most of the studied days the local shipping emissions have an ozone depletion effect at daytime while the local shipping VOC emission have ozone formation effect peaking in the morning and sometimes also in the afternoon (Fig. S5c). The regional shipping increases the ozone concentrations in all three depicted cases, showing maxima in the afternoon.

[Figure]

(d)

[Figure]

*Figure S5: (a) Modelled daily mean contributions to ozone concentrations from local shipping, regional shipping and VOCs emissions from local shipping at Eriksberg in summer (JJA) 2012. (b)–(d) Diurnal variation of the contributions to ozone concentration in panel (a) on selected days: (b) 2 June, 2012; (c) 7 July, 2012 and (d) 5 August, 2012.*

8.  *the section of 3.2 Impact of ship emissions on local air quality Most of this section simply describes the figure by using domain average, which does not make too much sense and not too much useful. Some comparisons might be made with either other sources or other studies to reveal the advancement of this study. For instance, what do the contributions from the local and regional shipping emission tell us? Is it useful in future strategies in the control policy? Only simple descriptions greatly discount the value of the study.*

Response:

We would like to thank the Reviewer for pointing out the need of a deeper analysis to improve the value of the study. The air quality discussion arises from the annual mean for the later discussion on health impact. But we agree that more interesting comparisons will reveal the advancement of the study. The section 3.2 has been updated from several aspects, discussing local relative and absolute contributions, seasonal differences and especially exemplifying more details of the impacts of shipping and other sources for location Eriksberg, as described in more detail bellow. Regarding the control policies we would like to refer to the Conclusions part of the paper where we discuss potential impact of different policies for mitigation of air pollution from shipping on exposure to air pollutants in Gothenburg and on the health effects.

Section 3.2.1 SO$_2$
The modelled SO$_2$ concentrations in Gothenburg are relatively low and Fig. 6 shows highest concentrations around the city ports as well as around industrial areas north of Göta älv.  The dominated south-westerly winds transport emissions from the shipping routes and port areas farther inlands to the north of Göta älv. Eriksberg, located on the north of Göta älv, is today a modern residential and commercial center built in place of an old dockyard area. We have selected this place to study relative impact of shipping in more detail. The shipping-related monthly contributions to SO$_2$ concentrations at Eriksberg were

47 % on average and over 60 % on June-August. Figure S3 in the supplement shows the modelled monthly mean relative contributions at Eriksberg.

[Figure]

*Figure S3: Modelled monthly mean relative contributions from local shipping, regional shipping and all other emission sources (road traffic, industry etc.) to SO₂ concentrations at Eriksberg in year 2012.*

Section 3.2.2 NO2

Nearly 90 % of $NO_x$ emissions in Gothenburg are from road traffic (47 %) and local shipping (41 %). But local shipping impact is concentrated in areas inside the harbor along the Göta älv and decreases with growing distance to the port areas. Fig. 8 presents the impacts of local, regional shipping, as well as road traffic and other local anthropogenic sources on monthly level at Eriksberg, located on the north of Göta älv. The modelled annual mean $NO_2$ concentration from all sources is 7.5 ppb at Eriksberg, in which 2.5 ppb (33 %) from local shipping, 1.0 ppb (13 %) from regional shipping and 2.1 ppb (28 %) from road traffic. The maximum relative contributions from local shipping and regional shipping to monthly mean concentrations of $NO_2$ reach to 43 % in July and 16 % in June respectively. The monthly average contributions from local and regional shipping together are larger than or comparable to the contributions from road traffic in all months. Even though road traffic is the major contributor to the $NO_2$ concentrations in urban environment, the local ship emissions should not be neglected, especially in areas close to the city ports.

[Figure]

*Figure 8: Modelled monthly mean contributions of the local shipping, regional shipping, local road traffic and other anthropogenic emissions (including contribution from the boundary conditions) to the NO2 concentrations (ppb) at Eriksberg in year 2012.*

Section 3.2.3 $O_3$
Major update already shown in question 7.

Section 3.2.4 Particulate matter
At the near-harbour residential area Eriksberg, the modelled annual mean $PM_{2.5}$ concentration from all sources is 4.5 µg m$^{-3}$. The calculated annual mean contributions from local shipping and regional shipping are 0.2 µg m$^{-3}$ (~4%) and 0.4 µg m$^{-3}$ (~9%) respectively. The maximum monthly relative contribution from the local and regional shipping was about 29 % in July, in which 21 % from regional shipping (Fig. 11). Road traffic, the largest local source of $PM_{10}$, contributed up to 5 % of monthly $PM_{2.5}$ mean concentrations. The large contribution of $PM_{2.5}$ from regional shipping is agree with the character of source apportionment in Gothenburg. An early study shows that the main sources types of $PM_{2.5}$ in Gothenburg were long-range transport (LRT) (about 50 %), followed by ship emissions (20 %) and local combustion (19 %) (Molnár et al., 2017).

[Figure]

*Figure 11: Modelled monthly mean contributions from local shipping, regional shipping and other sources (including contribution from the boundary condition) to PM$_{2.5}$ concentrations (µg m$^{-3}$) at Eriksberg for year 2012.*

***Minor comments:***

1. *Line 48 Our study show "show" changed to "shows"*
2. *Line 49: emphasising changed to emphasizing*
3. *Line 157: exposureresponse Please add a space between exposure and response*

Response: Thank you, corrected accordingly.

4. *Line 233: In this study, the meteorological component of TAPM was driven by the recently published ECMWF ERA5 synoptic Since the COSMO-CLM model has higher meteorological model and TAPM was driven by CMAQ 4km * 4km, why not using COSMO-CLM drives TAPM?*

Response:

We would like to thank the reviewer for this comment, we agree that use of the same meteorological driver for both models would make sense. Reason for using the two different meteorological drivers for the TAPM and CMAQ modelling has a practical background – the work was following two different tasks which were initiated by two different modelling teams. The use of common meteorological driver was not agreed, mainly as TAPM produces its own fields and the 2 modelling studies were connected through the boundary concentration fields. The model comparison is discussed in more detail in Point 5 of the Major comment section.

5. *Line 235: five nested domains What are the five domains? It is better to show a figure of the five nested domains. The authors also need to clarify the spatial resolutions of the five domains. Fig. 3b: the domain should be inferred in Fig. 3a, so the readers can tell where the domain of the finer resolution is.*

Response: The Fig. 3a has been updated accordingly.

[Figure]

*Figure 3: (a) Five nested meteorological model domains with their sizes and spatial resolutions. The fifth domain with air pollution grid (250 x 250 m2) is pointed out in the figure, showing location of the three air quality monitoring sites Femman, Haga and Mölndal, as well as Eriksberg, a residential are close to the harbour;*

6. *Line 307: "NOX" should be replaced by "NOx".*
7. *Line 512: "μg m-3" should be replaced by "μg m-3", and the same applies to Line 523 and 549.*
8. *Line 488: "A3 in Appendix", but the Appendix only have S3, not A3. The same issue applies to "Fig. A4 in the Appendix" on line 503 and 511.*

Response: Thank you, corrected accordingly.

9. *Fig 8, the x-axis label needs to be changed. For instance, either all using the mid-day of the month, i.e., 15/01, or something else to make it easy to follow.*

Response: Yes, using the mid-day of the month would be much easier to follow. However, the figure has been changed to monthly mean according to the suggestion of the first reviewer.

[Figure]

*Figure 8: Modelled monthly mean contributions of the local shipping, regional shipping, local road traffic and other anthropogenic emissions (including contribution from the boundary conditions) to the NO2 concentrations (ppb) at Eriksberg in year 2012.*

10. *Figure captions can be more succinct. A lot of repetitive words.*

Response: Corrected accordingly.

**References**

Haeger-Eugensson, M., Moldanova, J., Ferm, M., Jerksjö, M., Fridell, E.: On the increasing levels of NO2 in some cities. The role of primary emissions and shipping. IVL report B1886, www.ivl.se, 2010.

Hanna, S. and Chang, J.: Acceptance criteria for urban dispersion model evaluation. Meteorol. Atmos. Phys., 116, 133–146, doi:10.1007/s00703-011-0177-1, 2012.

Karl, M., Walker, S. E., Solberg, S., Ramacher, M. O. P.: The Eulerian urban dispersion model EPISODE – Part 2: Extensions to the source dispersion and photochemistry for EPISODE–CityChem v1.2 and its application to the city of Hamburg. Geosci. Model Dev., 12, 3357–3399, doi:10.5194/gmd-12-3357-2019, 2019.

Molnár, P., Tang, L., Sjöberg, K., Wichmann, J.: Long-range transport clusters and positive matrix factorization source apportionment for investigating transboundary PM2.5 in Gothenburg, Sweden. Environmental Science: Processes & Impacts 19, 1270-1277, 2017. DOI: 10.1039/C7EM00122C.

Nenes, A., Pandis, S.N., Pilinis, C.: ISORROPIA: A New Thermodynamic Equilibrium Model for Multiphase Multicomponent Inorganic Aerosols. *Aquatic Geochemistry* **4,** 123–152, 1998.

Pszeny, A.A.P., Moldanová, J., Keene, W.C., Sander R., Maben J.R., Martinez-Harder, M. Crutzen, P.J., Perner, D. and Prinn, R.G.: Aerosol pH and Inorganic Halogen Species in the Hawaiian Marine Boundary Layer, Atmos. Chem. Phys., 4, 147-168, 2004.

Ramacher, M. O. P., Karl, M.: Integrating Modes of Transport in a Dynamic Modelling Approach to Evaluate Population Exposure to Ambient $NO_2$ and $PM_{2.5}$ Pollution in Urban Areas. IJERPH, 17, 2099, doi:10.3390/ijerph17062099, 2020.

Ramacher, M. O. P., Karl, M., Bieser, J., Jalkanen, J.-P., and Johansson, L.: Urban population exposure to NOx emissions from local shipping in three Baltic Sea harbour cities – a generic approach, Atmos. Chem. Phys., 19, 9153–9179, doi:10.5194/acp-19-9153-2019, 2019.

Ramacher, M.O.P.: Performance and evaluation of local scale wind flow fields for urban air pollution modeling with the coupled prognostic model TAPM driven by ERA5 climate reanalysis data, abstract EGU2018-8112 accepted for poster presentation at the European Geosciences Union General Assembly, Vienna, 2018.

Sander, R., and Crutzen. P.J.: Model study indicating halogen activation and ozone destruction in polluted air masses transported to the sea. J. Geophys. Res., 101, 9121-9138, 1996.

Solazzo, E., Bianconi, R., Pirovano, G., Matthias, V., Vautard, R., Moran, M. D., Appel, K. W., Bessagnet, B., Brandt, J., Christensen, J. H., Chemel, C., Coll, I., Ferreira, J., Forkel, R., Francis, X. V., Grell, G., Grossi, P., Hansen, A. B., Miranda, A. I., Nopmongcol, U., Prank, M., Sartelet, K. N., Schaap, M., Silver, J. D., Sokhi, R. S., Vira, J., Werhahn, J., Wolke, R., Yarwood, G., Zhang, J., Rao, S. T. & Galmarini, S.: Operational model evaluation for particulate matter in Europe and North America in the context of AQMEII. *Atmospheric Environment*, 53, 75-92, 2012.

Tang, L., Miao, J.-F, Chen, D.: Performance of TAPM against MM5 at urban scale during GÖTE2001 campaign. Boreal Environment Research, 14, 338-350, 2009.